# PMark: Towards Robust and Distortion-free Semantic-level Watermarking with Channel Constraints

**Jiahao Huo**[1], **Shuliang Liu**[1,2], **Bin Wang**[3], **Junyan Zhang**[1,4], **Yibo Yan**[1,2], **Aiwei Liu**[5],
**Xuming Hu**[1,2*], **Mingxun Zhou**[2*]

[1]The Hong Kong University of Science and Technology (Guangzhou)
[2]The Hong Kong University of Science and Technology   [3]Peking University
[4]National University of Singapore   [5]Tsinghua University
jiahaohuotj@gmail.com mingxunz@ust.hk

## ABSTRACT

Semantic-level watermarking (SWM) for large language models (LLMs) enhances watermarking robustness against text modifications and paraphrasing attacks by treating the sentence as the fundamental unit. However, existing methods still lack strong theoretical guarantees of robustness, and reject-sampling–based generation often introduces significant distribution distortions compared with unwatermarked outputs. In this work, we introduce a new theoretical framework on SWM through the concept of proxy functions (PFs) – functions that map sentences to scalar values. Building on this framework, we propose PMark, a simple yet powerful SWM method that estimates the PF median for the next sentence dynamically through sampling while enforcing multiple PF constraints (which we call channels) to strengthen watermark evidence. Equipped with solid theoretical guarantees, PMark achieves the desired distortion-free property and improves the robustness against paraphrasing-style attacks. We also provide an empirically optimized version that further removes the requirement for dynamical median estimation for better sampling efficiency. Experimental results show that PMark consistently outperforms existing SWM baselines in both text quality and robustness, offering a more effective paradigm for detecting machine-generated text. The source code is available at this URL.

## 1 INTRODUCTION

The rapid advancements of generative AI (GenAI) techniques (Achiam et al., 2023; Team et al., 2025; Rombach et al., 2022; Liu et al., 2024; Brooks et al., 2024) have transformed content creation across diverse fields (Mittal et al., 2024; Chen et al., 2025; Yang et al., 2024), raising significant concerns regarding the traceability of AI-generated text and copyright protection (Zhao et al., 2024; Wu et al., 2025). Watermarking (Kirchenbauer et al., 2023; Aaronson, 2023), which embeds distinctive patterns into generated content, has emerged as a critical solution to these challenges.

Token-level watermarking schemes for text generation have been widely studied. The popular Green-Red scheme (Kirchenbauer et al., 2023; Zhao et al., 2023a) biases token sampling toward a "green" subset, enabling watermark detection via statistical tests on the frequency of green tokens. However, Green-Red watermarking schemes are inherently *biased* (Hu et al., 2023), meaning that they deviate from the original sampling distribution of LLMs and may degrade text quality (Hu et al., 2023). Distortion-free methods such as Gumbel Watermarking (Aaronson, 2023) and PRC-based schemes (Christ & Gunn, 2024) have also been explored, typically associated with cryptographic techniques such as digital signatures (Rivest et al., 1978). Recently, some studies (Tsur et al., 2025b;a) also discuss the best trade-off between detectability and text quality by maximizing the likelihood of watermark detection while minimizing the distortion of generated text.

Unfortunately, token-level watermarks can be easily removed: an attacker can simply ask an unwatermarked model to rephrase the generated text while preserving most of its semantic information (Hou et al., 2023). To improve robustness against such attacks, semantic-level watermarking (SWM) approaches like SemStamp (Hou et al., 2023; 2024) treat a semantically complete sentence as the fundamental watermarking unit. These methods employ rejection sampling to ensure that generated sentences fall within a valid semantic region of the embedding space, analogous to the green

---

*Co-corresponding authors: Mingxun Zhou and Xuming Hu.

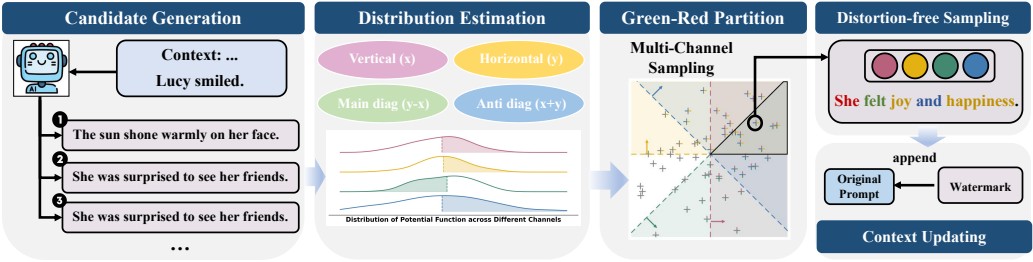

Figure 1: Illustration of PMARK pipeline in 2D space, with robustness enhanced by multi-channel constraints. Note that we use orthogonal pivots and distortion-free partition in practice.

list in Green-Red watermarking. Nonetheless, this split-and-reject paradigm inherits the distortion drawback of Green-Red watermarking and also introduces additional weaknesses, such as sampling failures when all candidate sentences fall in invalid regions (Zhao et al., 2024).

In this work, we propose a new semantic-level watermarking scheme named PMARK with the distortion-free property. To achieve this goal, we first establish a solid framework for analyzing SWM schemes based on a core concept called the *proxy function* (PF). Within this framework, PMARK defines the PF of a sentence as the cosine similarity between its embedding and a pre-defined random vector. The key idea behind PMARK is that, given knowledge of certain statistical properties of the PF distribution, we can quantify the probability mass of the valid region and perform a sampling process with a strong theoretical guarantee of being distortion-free. We further incorporate multiple channel constraints to enhance the density of watermarking evidence, as illustrated in Figure 1. Leveraging the fact that random vectors in high-dimensional spaces are almost always orthogonal (Ruppert, 2004), we show that even when simply estimating the median PF as zero, PMARK still achieves high text quality with an efficient sampling process.

Empirically, our evaluation demonstrates that while PMARK preserves high text quality (PPL 4.37–4.71), it achieves significantly better robustness against paraphrasing attacks using GPT Para-phraser (OpenAI, 2022), with improvements of up to 14.8% and 44.6% over the prior best semantic-level and token-level watermarking schemes, respectively. In terms of sampling efficiency, the online version of PMARK requires only 20% of the resources (measured in token consumption) compared to prior SOTA semantic-level watermarking schemes, while the offline version exhibits even lower consumption, paving the way for real-world deployment of PMARK in practice.

**Key contributions of this paper include:**

❶ We propose a novel theoretical framework that unifies existing SWMs via the introduction of the proxy function, providing solid analytical foundations for performance evaluation.

❷ We identify that sparse watermarking evidence in SWMs negatively impacts adversarial robustness, and address this problem by introducing multiple channel constraints.

❸ We introduce an online version of PMARK that achieves high robustness and is theoretically distortion free under mild prerequisites; we also present an offline version that reduces computational cost while maintaining low distortion.

❹ We conduct comprehensive experiments across a variety of text watermarking tasks across multiple datasets and backbones, validating the effectiveness and efficiency of our method.

## 2 PRELIMINARIES

In this section, we delve into the problem of Semantic-level Watermarking (SWM), which treats semantically complete sentences as the fundamental unit. We begin by formalizing the problem and defining key concepts, inspired by Hu et al. (2023). See Appendix B for detailed proofs.

**Zero- and Multi-bit Watermarks.** We write $\mathcal{M}(\pi) \rightarrow y$ to denote the process of sampling a response $y$ from the model given an input prompt $\pi$. A watermark generation algorithm is denoted as Generation$_k^{\mathcal{M}}(\pi) \rightarrow y$, where $k$ is the watermark generation key (Zhao et al., 2024). A detection algorithm determines whether a given text $y$ is watermarked, denoted as Detect$(y) \rightarrow \{\text{True}, \text{False}\}$. In this work, we focus on *zero-bit* watermarks, which embed a single detectable signal (e.g., True or False) into text. In contrast, the concurrent **SAEMark** (Yu et al., 2025) investigates *multi-bit* watermarking, which embeds a message m $\in \{0,1\}^m$ into text (Lau et al., 2024). A detailed discussion of concurrent studies and related work can be found in Appendix C.

**Problem Setup.** Let $\Sigma$ denote the vocabulary set of LLM. We define $\Sigma^*$ as the set of all semantically complete sentences, including the null sentence of length zero. For any $s \in \Sigma^*$, let $\overline{s}$ denote its token sequence, $\overline{s} = [x_1^s, \ldots, x_{|s|}^s] = \text{tokenize}(s)$.

At each generation step, the probability of producing the next token $x_{n+1} \in \Sigma$ given the current context $x_1, \ldots, x_n$ is denoted by $P_M(x_{n+1} \mid x_1, \ldots, x_n)$. The joint probability of generating a sequence of $m$ tokens $x_{n+1}, \ldots, x_{n+m}$ is expressed as:

$$P_M(x_{n+1}, \ldots, x_{n+m} \mid x_1, \ldots, x_n) = \prod_{i=1}^{m} P_M(x_{n+i} \mid x_1, \ldots, x_n, x_{n+1}, \ldots, x_{n+i-1}). \quad (1)$$

In the context of SWM, we focus on the probability distribution of the next sentence $s_{n+1} \in \Sigma^*$ given the preceding context $s_1, \ldots, s_n$, denoted as:

$$P_M(s_{n+1} \mid s_1, \ldots, s_n) = \prod_{i=1}^{|s|} P_M(x_i^s \mid \overline{s_1}, \ldots, \overline{s_n}, x_1^s, \ldots, x_{i-1}^s). \quad (2)$$

Here $P_M(s) : \Sigma^* \to [0, 1]$ is a probability mass function over the countable set $\Sigma^*$.

For watermarking, we introduce a private key $k$ drawn from a key space $K$, where $k \in K$ is randomly sampled from a prior distribution $P_K(k)$. The watermarked output of the LLM follows a conditional distribution $P_M^w(s_{n+1} \mid s_1, \ldots, s_n; k)$, which depends on the preceding context and $k$.

**Sampling-based Semantic-level Watermarking.** Directly modeling $P_M(s_{n+1} \mid s_1, \ldots, s_n)$ is generally intractable since $\Sigma^*$ is infinite. A common approach adopted by SWM methods is to sample a set of i.i.d. candidate sentences $x_1, \ldots, x_N \sim P_M(s_{n+1} \mid s_1, \ldots, s_n)$, and then select a candidate $Y$ that carries watermark evidence, $Y_k = \text{SWM}_k(\{x_1, \ldots, x_N\})$. The distribution of the watermarked output is then defined as the distribution of $Y$:

$$Y_k \sim P_M^w(s_{n+1} \mid s_1, \ldots, s_n; k). \quad (3)$$

**Definition 1** (**Single-sentence Distortion-Free**). *Given a context* $\boldsymbol{\pi} = [s_1, \ldots, s_n]$, *we say that the watermarked LLM distribution is* single-sentence distortion-free *relative to the original LLM* $P_M$ *if, for all* $s_{n+1} \in \Sigma^*$,

$$P_M(s_{n+1} \mid \boldsymbol{\pi}) = \sum_{k \in K} P_K(k) \cdot P_M^w(s_{n+1} \mid \boldsymbol{\pi}; k). \quad (4)$$

This condition means that the marginal distribution of $s_{n+1}$, when averaged over all possible watermark keys $k$, is identical to its unwatermarked distribution under the original model.

**Definition 2** (Equivalence via Watermark Code Space). *In most cases,* $P_K(k)$ *is designed to induce uniform random sampling over a watermark code space* $E$, *where* $e_K \sim \text{Uniform}(E)$. *Under this construction, the right side of equation 4 can be equivalently rewritten as:*

$$\sum_{k \in K} P_K(k) \cdot P_M^w(s_{n+1} \mid \boldsymbol{\pi}; k) = \sum_{e \in E} P(e_K = e) \cdot P_M^w(s_{n+1} \mid \boldsymbol{\pi}; e). \quad (5)$$

**Definition 3** (Probability Measure on $\Sigma^*$). *Let* $(\Sigma^*, \mathcal{G}, \mu_M)$ *be a probability space, where* $\mathcal{G}$ *is a* $\sigma$-algebra over $\Sigma^*$, *and* $\mu_M$ *is the probability measure induced by the model* $P_M : \Sigma^* \to [0, 1]$. *For any measurable subset* $A \subseteq \Sigma^*$, *the measure* $\mu_M(A)$ *is defined as*

$$\mu_M(A) = \sum_{s \in A} P_M(s).$$

# 3 REVISITING SEMANTIC-LEVEL WATERMARKING

## 3.1 PROXY FUNCTION

Section 2 discusses the sampling-based paradigm of SWM, which addresses the challenge of operating over the infinite set $\Sigma^*$. However, to enable rule-based selection among a candidate sentence set $x_1, \ldots, x_N \in \Sigma^*$, SWM methods require a concrete mapping from the sentence space to the real numbers, since direct computation in the infinite textual space is generally intractable.

**Proxy Function.** To this end, we define a function $\mathcal{F} : \Sigma^* \to \mathbb{R}$, referred to as the *proxy function*, which assigns a real-valued score to each semantically complete sentence. Let $U$ denote the range of the proxy function $\mathcal{F}$. We define a partition of the range $U$ as $\mathcal{P}(U)$ (see Definition 4). Each element $u \in U$ belongs to exactly one subset $A_i \in \mathcal{P}(U)$, denoted as $A_i = \mathcal{P}_u(U)$.

We also define the inverse mapping of $\mathcal{F}$, denoted by $\mathcal{F}^{-1} : U \to 2^{\Sigma^*}$, where $2^{\Sigma^*}$ is the power set of $\Sigma^*$. Each value $u \in U$ thus maps to the set of sentences in $\Sigma^*$ that share the same proxy value under

$\mathcal{F}$, denoted as $\mathcal{F}^{-1}(u)$. Furthermore, the watermark code space is denoted as $E \in 2^{\mathcal{P}(U)}$, whose element $e$ is selected by the watermarking scheme according to a prior distribution $e_K \sim P_K(E)$ induced by the key $P_K(k)$.

## 3.2 Analysis of the Baseline

**Proxy Function of SemStamp.** We first focus on a simplified generation process of SemStamp, where the original margin constraint is removed for clarity. In SemStamp (Hou et al., 2023), the proxy function is defined as:

$$\mathcal{F}(s) = \text{LSH}(\mathcal{T}(s)) = [\text{LSH}_1(\mathcal{T}(s))\| \cdots \|\text{LSH}_h(\mathcal{T}(s))], \tag{6}$$

where $\text{LSH}_i(v) = \text{sign}(t_i \cdot v)$ (Indyk & Motwani, 1998; Charikar, 2002), $\{t_i \mid i = 1, \dots, h\} \subset \mathbb{R}^d$ is a set of randomly initialized vectors, $\mathcal{T}$ is a text encoder with $\mathcal{T}(s) : \Sigma^* \to \mathbb{R}^d$, and $[\cdot]$ denotes the transformation from binary representation to its corresponding decimal value. Given that the range of $\mathcal{F}$ is $U = \{1, \dots, 2^h\}$, the partition used in SemStamp is defined as $\mathcal{P}(U) = \{\{1\}, \dots, \{2^h\}\}$.

**Sampling-then-selecting Process.** We now reformulate the rejection-sampling process of Sem-Stamp as a sampling-then-selecting process, without loss of generality:

- Given a context $\boldsymbol{\pi}$, SemStamp samples $N$ candidate sentences from the natural distribution: $W = \{x_1, \dots, x_N\}, x_i \sim P(s \mid \boldsymbol{\pi}), \quad i = 1, \dots, N$.

- For a fixed green ratio $\gamma$ such that $\gamma|\mathcal{P}(U)| \in \mathbb{N}_+$, the watermark code space is defined as $E = \{e \in 2^{\mathcal{P}(U)} \mid |e| = \gamma|\mathcal{P}(U)|\}$. The watermark code $e_K$ is sampled uniformly at random from $e_K \sim \text{Un}(E)$, implying that each element of $\mathcal{P}(U)$ is selected with probability $\gamma$. The *green region* of $\Sigma^*$ is then defined as $G = \{s \in \Sigma^* \mid \mathcal{F}(s) \in \bigcup e\}$, while the set of valid candidates is $V = \{x_i \mid \mathcal{F}(x_i) \in \bigcup e\}$. We assume that $V$ is always non-empty.

- A watermarked sentence $Y$ is then uniformly selected from $V$, where the distribution of the watermarked output is $Y \sim P_M^w(s \mid \boldsymbol{\pi})$.

**Distortionary Distribution of Watermarked Output.** Fix a context $\boldsymbol{\pi}$ and let $\mathcal{F} : \Sigma^* \to U$ be the proxy function with a finite range $U$ of size $M \triangleq |U|$. For $\forall u \in U$, define its natural mass:

$$q(u) \triangleq \mu_M(\mathcal{F}^{-1}(u)) = \sum_{t \in \mathcal{F}^{-1}(u)} P_M(t \mid \boldsymbol{\pi}), \qquad \sum_{u \in U} q(u) = 1. \tag{7}$$

Fix a green ratio $\gamma \in (0, 1)$ with $g = \gamma M \in \mathbb{N}_+$. The code space consists of all size-$m$ subsets $S \subseteq U$, each chosen uniformly at random. For any $S \subseteq U$, define the green mass $q(S) \triangleq \sum_{v \in S} q(v)$.

**Lemma 1** (Probability Scaling of Green Region). *For any fixed green set $S \subseteq U$ and any $N \geq 1$, the distribution of the output sentence $Y$ is independent of $N$ and equals the natural distribution conditioned on $\mathcal{F} \in S$. Specifically, for any $s \in \Sigma^*$ with $u = \mathcal{F}(s)$,*

$$P(Y = s \mid \boldsymbol{\pi}, S) = \begin{cases} \dfrac{P_M(s \mid \boldsymbol{\pi})}{q(S)}, & u \in S, \\ 0, & u \notin S. \end{cases} \tag{8}$$

Averaging over the uniformly random choice of green set $S$ yields a closed-form expression:

**Theorem 2** (Closed-form of Watermarked PMF). *For any $s \in \Sigma^*$ with $u = \mathcal{F}(s) \in U$,*

$$P_M^w(s \mid \boldsymbol{\pi}) = P_M(s \mid \boldsymbol{\pi}) \cdot \frac{1}{\binom{M}{g}} \sum_{\substack{S \subseteq U \\ |S|=m, u \in S}} \frac{1}{q(S)}. \tag{9}$$

*where $q(u)$ is defined in equation 7, and $M, g, u, U, \Sigma^*, \mathcal{F}, \mathcal{F}^{-1}$ are known quantities.*

**Corollary 2.1.** *$P_M^w(s \mid \boldsymbol{\pi})$ is distortion-free if and only if $q(u) = \frac{1}{M}$ for all $u \in U$.*

However, the condition in Corollary 2.1 is not guaranteed in SemStamp, whose embedding space is randomly partitioned. Similar conclusions apply to other SWM methods such as k-SemStamp and SimMark, with further analysis provided in Appendix D.1 and E.

## 3.3 Single-channel Distortion-free Sampling

In this section, we present a toy example of a distortion-free sampling scheme for next-sentence generation, serving as a foundation for subsequent sections.

**Proxy Function via Pivot Vector.** Given a text encoder $\mathcal{T} : \Sigma^* \to \mathbb{R}^d$ and a fixed pivot vector $v \in \mathbb{R}^d$, we define the proxy function as $\mathcal{F}_v(s) = \langle v, \mathcal{T}(s) \rangle = \frac{v \cdot \mathcal{T}(s)}{\|v\| \cdot \|\mathcal{T}(s)\|}$, which denotes the cosine similarity between the pivot vector and the encoded sentence. The range of $\mathcal{F}_v$ is $[-1, 1]$.

**Median-based Sampling.** Given a context $\boldsymbol{\pi}$, we sample $N$ i.i.d. candidate sentences from the natural distribution: $W = \{x_1, \ldots, x_N\}, x_i \sim P_M(s \mid \boldsymbol{\pi})$, where $N$ is an even integer. The proxy function $\mathcal{F}_v$ assigns each candidate a scalar score $F = \{f_i = \mathcal{F}_v(x_i) \mid i = 1, \ldots, N\}$. Let $m_v$ be the median of $F$. We partition $W$ into two equal-sized subsets:

$$F_{\text{upper}} = \{x_i \mid f_i \geq m_v\}, \quad F_{\text{lower}} = \{x_i \mid f_i < m_v\}, \tag{10}$$

with ties broken randomly to ensure balance.

Define the watermark code space as $E = \{F_{\text{upper}}, F_{\text{lower}}\}$. A random seed $k \in \{0, 1\}$ is sampled

uniformly, and the selected code is $e_K = \begin{cases} F_{\text{upper}}, & \text{if } k = 1, \\ F_{\text{lower}}, & \text{if } k = 0. \end{cases}$

The valid sentence set is $V = e_K$, and the next sentence is sampled uniformly from $V$: $Y \sim \text{Un}(V)$. We denote the operation that selects the half-partition $V$ from the whole set $W$ based on random seed $k$ and proxy function $\mathcal{F}_v$ as $V = \mathcal{D}(W \mid k, \mathcal{F}_v), |V| = |W|/2$.

**Theorem 3** (Distortion-free on a Single Channel). *For any context $\boldsymbol{\pi}$, sample $N$ i.i.d. candidates $W = \{x_1, \ldots, x_N\}$ from $P_M(\cdot \mid \boldsymbol{\pi})$, and form half-sets $e_0, e_1$ by the median rule. Draw $k \sim \text{Unif}\{0, 1\}$ and then sample $Y$ uniformly from $e_k$. Then $P_M^w(s \mid \boldsymbol{\pi}) = P_M(s \mid \boldsymbol{\pi})$.*

*Proof.* For any $x_i \in W$, we have

$$P(Y = x_i \mid W) = \sum_{k \in \{0,1\}} P(k) P(Y = x_i \mid W, k) = \frac{1}{2} \left( \frac{2}{N} \mathbf{1}\{x_i \in e_0\} + \frac{2}{N} \mathbf{1}\{x_i \in e_1\} \right) = \frac{1}{N}.$$

Averaging over the randomness of $W$ yields

$$P_M^w(s \mid \boldsymbol{\pi}) = \sum_{i=1}^{N} P(x_i = s) P(Y = x_i \mid W) = P_M(s \mid \boldsymbol{\pi}),$$

which proves that the single-channel sampling method above is distortion-free. $\qquad\square$

## 4 PROPOSED METHOD

### 4.1 MULTI-CHANNEL CONSTRAINED GENERATION (ONLINE)

**Single-Channel Robustness.** While Section 3.3 introduces a simple distortion-free sampling strategy for SWM, it remains vulnerable to paraphrasing attacks.

**Theorem 4** (Semantic Robustness on Single Channel). *Let $s$ be a watermarked sentence and $v \in \mathbb{R}^d$ the pivot vector. Suppose an attacker $\mathcal{A} : \Sigma^* \to \Sigma^*$ modifies $s$ such that $1 - \langle \mathcal{T}(\mathcal{A}(s)), \mathcal{T}(s) \rangle \leq d$. Then the probability that the watermark evidence of $s$ is removed by such an attack is bounded by*

$$P_{\text{rm}} \leq \mu_M \left( s \in \Sigma^* \Big| \mathcal{F}_v(s) \in [m_v - \sqrt{2d}, m_v + \sqrt{2d}] \right). \tag{11}$$

The robustness of the entire watermarked text $S = [s_1, \ldots, s_n]$ depends on the specific detection scheme, with a detailed analysis available in Giboulot & Furon (2024). Importantly, compared with token-level baselines where each token contributes one bit of watermark evidence, the evidence density of SWM in the single-channel setting is extremely sparse. This sparsity results in a significant degradation in detectability under adversarial attacks. To address this limitation, we extend the paradigm to incorporate multiple channel constraints.

**Multi-Channel Sampling.** Given a pre-defined set of $b$ orthogonal vectors $v_1, \cdots, v_b$, we refer to each pivot as a *channel*. The proxy function defined on $v_i$ is $\mathcal{F}_{v_i}(s) = \langle v_i, \mathcal{T}(s) \rangle$, also denoted as $\mathcal{F}_i$ for simplicity. Specifically, these pivots can be generated from the QR decomposition of a matrix. Since this approach relies on dynamic median estimation, we refer to it as the online PMARK.

1. Given a context $\boldsymbol{\pi}$, PMARK samples $N$ candidate sentences from the natural distribution: $W = \{x_1, \ldots, x_N\}, x_i \sim P(s \mid \boldsymbol{\pi}), i = 1, \ldots, N$.

2. For $i = 1$ to $b$, update $V_i = \mathcal{D}(V_{i-1} \mid k_i, \mathcal{F}_i)$, where $V_0 = W$ and $k_i \sim \text{Un}(\{0, 1\})$.

3. A watermarked sentence $Y$ is then uniformly sampled from $V_b$, so that the distribution of the watermarked output is $Y \sim P_M^w(s \mid \boldsymbol{\pi})$.

**Algorithm 1:**
PMARK Online Generation

**Input** : LLM $M$; prompt $s^{(0)}$;
   encoder $\mathcal{T}$; $T$; $b$;
   random seeds $R = \{r_{(t,j)}\}$;
   sample budget $N$;
**Output:** $s^{(1)}, \ldots, s^{(T)}$;
1 Initialize $b$ orthogonal pivots
   $v^{(1)}, \ldots, v^{(b)}$;
2 **for** $t \leftarrow 1$ **to** $T$ **do**
3    Sample $W^{(t)} = \{x^{(1)}, \ldots, x^{(N)}\}$
      with $x^{(i)} \sim P_M(s \mid s^{(0:t-1)})$;
4    $V^{(0)} \leftarrow W^{(t)}$;
5    **for** $j \leftarrow 1$ **to** $b$ **do**
6      $\mathcal{F}_j(s) = \langle v^{(j)}, \mathcal{T}(s) \rangle$;
7      $V^{(j)} \leftarrow$
        $\mathcal{D}\big(V^{(j-1)} \mid r_{(t,j)}, \mathcal{F}_j\big)$ on
        channel $v^{(j)}$;
8    **end**
9    Select $s^{(t)} \sim \mathrm{Un}\big(V^{(b)}\big)$;
10 **end**
11 **return** $s^{(1)}, \ldots, s^{(T)}$;

**Algorithm 2:** PMARK Online Detection

**Input** : $S = [s^{(0)}, \ldots, s^{(T)}]$; $M$; $\mathcal{T}$; $v^{(1\ldots b)}$; $R = \{r_{(t,j)}\}$;
   $N$; $\alpha$; threshold $\delta$; Smooth Factor $K$
**Output:** True or False
1 $N_g \leftarrow 0$;   $N_{\text{total}} \leftarrow b \cdot T$;
2 **for** $t \leftarrow 1$ **to** $T$ **do**
3    Resample $W'_t = \{x^{(1)}, \ldots, x^{(N)}\}$ with
      $x^{(i)} \sim P_M(s \mid s^{(0:t-1)})$;
4    **for** $j \leftarrow 1$ **to** $b$ **do**
5      Define $\mathcal{F}_j(s) = \langle v^{(j)}, \mathcal{T}(s) \rangle$;
        $\widehat{m}_{(t,j)} \leftarrow \mathrm{HDMedian}\big(\{\mathcal{F}_j(x^{(i)})\}_{i=1}^N\big)$;
6      $x_{(t,j)} \leftarrow \mathcal{F}_j(s^{(t)})$;
7      **if** $r_{(t,j)} = 1$ **and** $x_{(t,j)} > \widehat{m}_{(t,j)} - \delta$ **then**
8        $c_{(t,j)} \leftarrow 1$
9      **else if** $r_{(t,j)} = 0$ **and** $x_{(t,j)} < \widehat{m}_{(t,j)} + \delta$ **then**
10        $c_{(t,j)} \leftarrow 1$
11      **else**
12        $c_{(t,j)} \leftarrow \exp(-K|x_{(t,j)} - \widehat{m}_{(t,j)}|)$
13      **end**
14      $N_g \leftarrow N_g + c_{(t,j)}$;
15    **end**
16 **end**
17 $z \leftarrow \dfrac{|N_g - 0.5\,N_{\text{total}}|}{\sqrt{0.25\,N_{\text{total}}}}$;    **return** ($z > z_\alpha$);

Figure 2: PMARK Online Watermarking. **Left**: Multi-channel constrained generation; **Right**: Detection via soft-$z$-test.

**Remark 1** (Multi-channel Distortion-free). *It is straightforward to verify that such multi-channel sampling is distortion-free, since for $\forall x_j \in V_i$, $P(x_j \in V_{i+1}) = \frac{1}{2}$ and $P(Y = x \mid x \in W) = 1/N$.*

The complete process for generating watermarked texts is described in Algorithm 1. A theoretical analysis outlining the advantages of multi-channel sampling is presented in Appendix B.

### 4.2 SOFT-$z$-TEST DETECTION (ONLINE)

We now introduce a robust detection scheme for text watermarked with the online PMARK.

1. **Sentence Segmentation.** The input text is segmented into sentences $S = [s_0, s_1, \ldots, s_T]$ to enable subsequent processing.

2. **Median Reconstruction via Sampling.** For each sentence $s_t$ ($1 \leq t \leq T$), resample $W'_t = \{x_1, \ldots, x_N\}$ i.i.d. from $P_M(s_t \mid s_{0:t-1})$.

3. **Online Median Estimation.** For each pivot vector $v_j$ ($1 \leq j \leq b$), estimate the median $m'_{(t,j)}$ of the values $\mathcal{F}_j(W'_t)$ using the Harrell–Davis estimator (Harrell & Davis, 1982).

4. **Smooth Counting of Watermark Evidence.** To mitigate discrepancies between the medians estimated during generation and detection, we introduce a smooth counting mechanism inspired by (Dabiriaghdam & Wang, 2025). Specifically, given a margin threshold $\delta > 0$ and a smoothing factor $K > 0$, the watermark evidence contributed by $s_t$ along axis $v_j$ is defined as
$$c_{(t,j)} = \begin{cases} 1, \text{if } (r_{(t,j)} = 1 \land \mathcal{F}_j(s_t) > m'_{(t,j)} - \delta) \lor (r_{(t,j)} = 0 \land \mathcal{F}_j(s_t) < m'_{(t,j)} + \delta), \\ e^{-K|\mathcal{F}_j(s_t) - m'_{(t,j)}|}, \text{otherwise.} \end{cases}$$

5. **Robust Detection via Soft-$z$-Test.** Finally, to determine whether the input text was generated by PMARK, we apply a soft-count-based $z$-test. The test statistic is computed as $z = \dfrac{\left|\sum_{t=1}^T \sum_{j=1}^b c_{(t,j)} - 0.5bT\right|}{\sqrt{bT \cdot 0.5 \cdot 0.5}}$, where $b \cdot T$ is the total number of potential watermark bits across channels.

The complete description of the online detection scheme is provided in Algorithm 2.

### 4.3 OFFLINE PARTITION WITH A PRIOR THRESHOLD

**Concentration of Measure.** In practice, we find the range of $\mathcal{F}$ is confined to a small interval around zero, $[-\epsilon, \epsilon]$ with $\epsilon \leq 0.08$ in most cases. This observation is consistent with Lemma 5:

**Lemma 5** (Orthogonality). *Let $x, y$ be random vectors drawn uniformly from the unit sphere $S^{d-1} \subset \mathbb{R}^d$. Then the angle $\theta$ between them has density*

$$p_d(\theta) = \frac{\Gamma\left(\frac{d}{2}\right)}{\Gamma\left(\frac{d-1}{2}\right)\sqrt{\pi}} \sin^{d-2}\theta \quad (\theta \in [0, \pi]). \tag{12}$$

Moreover, we observe a concentration phenomenon whereby the median $m$ remains close to zero. Building on this intuition and supporting empirical results, we use zero as an effective prior for the median in the offline PMARK. Experimental results in Section 5 demonstrate the effectiveness of this approximation. A brief analysis of the theoretical impact of this prior assumption is included in Appendix F.

**Offline Watermarking with a Prior-Median Assumption.** Accordingly, we adopt zero as the prior median during both generation and detection, while maintaining the same watermarking rate $\gamma$ for the $z$-test. For a candidate sentence $x$, define the binary signal vector

$$\text{Sig}(x) = \big[[f_1(x) > 0], \dots, [f_b(x) > 0]\big], \tag{13}$$

and let $r \in \{0, 1\}^b$ denote the random channel seeds. The total watermark evidence contributed by $x$ is

$$\text{E}(x) = \sum_{j=1}^{b} \big(\text{Sig}(x)(j) \wedge r(j)\big). \tag{14}$$

During generation, we select the sentence that maximizes this watermark evidence. This avoids the computational overhead of repeated sampling and the need to access the original prompt during detection by eliminating dynamical estimation, meaning that the offline detector will function without querying generator. The complete algorithm is also provided in Appendix F.

## 5 EXPERIMENTS

In this section, we conduct extensive experiments to answer the following research questions: (**RQ1**) How does PMARK perform under various attacks? (**RQ2**) Can PMARK enable high-quality text generation in practice? (**RQ3**) What is the computational cost of PMARK compared with other SWM baselines? (**RQ4**) How sensitive is PMARK to its key components or parameters?

### 5.1 EXPERIMENT SETUP

**Dataset and Baselines.** Following previous work (Hou et al., 2023; Wang et al., 2025), we evaluate the performance of PMARK using 500 samples from the C4 (Raffel et al., 2020) and BOOK-SUM (Kryściński et al., 2021) datasets separately, with OPT-1.3B (Zhang et al., 2022) and Mistral-7B-v0.1 (Jiang et al., 2023) as backbone models. Our baselines are based on the official implementation of MarkLLM (Pan et al., 2024), including token-level watermarking methods KGW (Kirchenbauer et al., 2023), UPV (Liu et al., 2023a), MorphMark (Wang et al., 2025), SIR (Liu et al., 2023b), EXP (Aaronson, 2023), EXPGumbel (Aaronson, 2023), SynthID (Dathathri et al., 2024), as well as semantic-level baselines SemStamp (Hou et al., 2023), k-SemStamp (Hou et al., 2024), and Sim-Mark (Dabiriaghdam & Wang, 2025). For the PMARK generation process, we set the number of channels to 4 with a sample budget $N = 64$. The hyperparameters of the soft-$z$-test during detection are set as $K = 150$ and $\delta = 0.001$. More implementation details can be found in Appendix G.1.

**Metrics.** Following previous work, we use perplexity (PPL) to measure the quality of generated text, computed using LLaMA-2-7B (Touvron et al., 2023). We also assess watermark effectiveness in terms of detectability (TPR@1%, AUC) and robustness under sentence-level attacks (Paraphrase Attack by Parrot (Sadasivan et al., 2023), GPT-3.5-turbo (OpenAI, 2022), and Back-Translation Attack by LLAMA-3.1-8B (Pan et al., 2024; Dubey et al., 2024)). Word-level attacks (Pan et al., 2024) such as word deletion and synonym substitution at rates 0.05, 0.15, and 0.30 are also considered. Refer to Appendix G.2 for a more detailed discussion of these attacks.

### 5.2 ROBUSTNESS AND TEXT QUALITY (RQ1 & RQ2)

To answer **RQ1**, we comprehensively compare PMARK with seven widely used token-level watermarks and three semantic-level watermarks under various attacks. Table 1 presents comprehensive results of watermarking methods under paraphrase attacks, with Pegasus (Pu et al., 2023), Parrot (Sadasivan et al., 2023), and GPT-3.5-turbo (OpenAI, 2022) as different paraphrasers. Figure 3 illustrates the trade-off between text quality and robustness under back-translation attacks, while Figure 4 shows the results of various SWM methods under word-level attacks. We provide the following observations (**Obs.**):

Table 1: Overall results for baseline methods and PMARK on OPT-1.3B and Mistral-7B across the BOOKSUM and C4 benchmarks. **Doc-P** denotes Paraphrase Attack, while Pegasus, Parrot, GPT are different paraphrasers. For each attack, we report TP@FP=1%, TP@FP=5%, and AUC, followed by the standard deviation of TP@FP=1% in light gray. **Bold** denotes the best result, and underlined denotes the second-best.

| Method | BOOKSUM | | | | C4 | | | |
|---|---|---|---|---|---|---|---|---|
| | No Attack↑ | Doc-P (Pegasus)↑ | Doc-P (Parrot)↑ | Doc-P (GPT)↑ | No Attack↑ | Doc-P (Pegasus)↑ | Doc-P (Parrot)↑ | Doc-P (GPT)↑ |
| **OPT-1.3B** | | | | | | | | |
| EXP (2023) | 98.8/99.0/99.3 | 4.2/14.6/52.5 | 5.4/16.0/55.2 | 4.4/13.4/53.0 | 99.0/99.6/99.8 | 54.6/68.2/86.9 | 42.4/59.6/84.6 | 21.2/37.2/73.3 |
| EXPGumbel (2023) | 99.4/99.4/99.4 | 12.4/20.2/50.6 | 12.4/20.0/52.2 | 13.0/23.2/54.4 | 98.6/98.6/99.3 | 75.2/84.6/91.8 | 74.6/81.0/92.8 | 55.6/66.0/86.0 |
| KGW (2023) | 100.0/100.0/100.0 | 1.4/4.6/56.8 | 1.4/5.9/55.7 | 0.6/4.2/54.6 | 100.0/100.0/100.0 | 89.3/97.0/99.2 | 76.2/91.4/98.1 | 51.4/78.5/95.0 |
| SIR (2023b) | 99.6/100.0/99.8 | 84.2/93.8/97.5 | 80.6/93.8/98.1 | 49.8/82.8/95.9 | 99.8/100.0/99.9 | 91.6/94.6/98.7 | 83.4/90.6/97.6 | 74.2/88.6/97.7 |
| UPV (2023a) | 100.0/100.0/100.0 | 90.0/95.6/98.7 | 89.2/98.2/99.3 | 73.6/91.0/98.0 | 100.0/100.0/100.0 | 90.6/95.4/98.3 | 78.7/88.4/97.4 | 80.9/91.3/98.2 |
| SynthID (2024) | 100.0/100.0/99.9 | 38.2/56.6/86.9 | 25.8/43.6/83.8 | 1.8/14.6/65.7 | 100.0/100.0/99.9 | 45.4/66.2/89.5 | 26.4/49.4/82.7 | 6.4/21.2/67.6 |
| MorphMark (2025) | 100.0/100.0/100.0 | 1.8/7.7/55.2 | 1.2/6.1/54.6 | 0.8/4.2/51.1 | 100.0/100.0/100.0 | 78.6/93.0/97.8 | 70.2/87.1/96.8 | 46.7/76.2/93.2 |
| SemStamp (2023) | 97.7/98.8/99.4 | 89.0/93.2/97.3 | 90.7/93.0/97.5 | 79.6/85.4/94.2 | 94.6/96.1/98.9 | 85.8/91.0/96.9 | 84.4/89.3/96.2 | 73.5/81.4/93.0 |
| k-SemStamp (2024) | 99.6/100.0/99.9 | 74.5/84.6/97.4 | 72.3/86.6/97.5 | 62.1/72.9/95.2 | 100.0/100.0/99.9 | 76.6/89.2/97.1 | 74.3/87.0/97.5 | 62.9/78.6/94.1 |
| SimMark (2025) | 88.2/94.0/98.8 | 23.4/42.8/84.6 | 29.0/45.6/87.3 | 22.0/40.4/83.6 | 77.6/94.2/98.5 | 11.6/35.8/82.9 | 13.6/39.2/84.5 | 16.0/44.8/86.5 |
| Ours(Online) | 99.8/99.8/99.9 | 97.4/99.0/99.8 | 96.8/99.0/99.8 | 95.4/99.0/99.6 | 100.0/100.0/99.9 | 96.2/99.6/99.8 | 97.2/99.0/99.8 | 97.8/99.6/99.8 |
| Ours(Offline) | 99.4/99.6/99.8 | 93.2/98.2/99.5 | 95.2/98.6/99.5 | 94.2/98.8/99.6 | 98.0/99.0/99.8 | 91.7/94.4/98.8 | 91.4/95.2/99.0 | 92.6/96.2/99.1 |
| **Mistral-7B** | | | | | | | | |
| EXP (2023) | 99.8/99.8/99.9 | 36.4/53.0/85.8 | 47.2/61.8/89.1 | 10.4/22.0/71.7 | 99.2/99.2/99.5 | 44.4/60.4/85.7 | 34.2/54.8/84.2 | 17.4/29.8/72.1 |
| EXPGumbel (2023) | 99.6/99.6/99.7 | 65.6/77.2/90.8 | 78.8/85.2/92.8 | 38.4/53.2/81.9 | 98.6/98.6/99.3 | 71.2/82.0/92.5 | 66.2/77.2/90.8 | 40.2/55.4/82.5 |
| KGW (2023) | 100.0/100.0/100.0 | 84.2/95.4/98.7 | 90.2/97.3/99.2 | 35.8/68.9/92.1 | 100.0/100.0/100.0 | 85.6/95.5/98.8 | 79.2/93.7/98.8 | 54.2/76.7/95.6 |
| SIR (2023b) | 100.0/100.0/99.9 | 85.0/92.6/98.0 | 80.8/92.0/98.2 | 41.2/71.4/93.6 | 100.0/100.0/99.9 | 87.6/93.4/98.1 | 83.2/89.6/97.8 | 63.4/80.8/96.4 |
| UPV (2023a) | 99.6/100.0/100.0 | 72.6/96.2/98.5 | 73.0/96.9/99.0 | 33.4/76.2/94.6 | 99.4/99.8/99.9 | 71.4/91.4/97.7 | 61.9/85.4/97.5 | 34.9/68.6/93.4 |
| SynthID (2024) | 100.0/100.0/100.0 | 30.2/49.2/83.5 | 28.2/48.4/85.5 | 3.0/13.0/65.0 | 99.8/99.8/99.8 | 47.0/56.0/87.6 | 31.4/41.4/80.7 | 7.4/17.2/68.2 |
| MorphMark (2025) | 99.8/100.0/100.0 | 68.2/88.9/97.2 | 77.4/93.3/98.4 | 24.6/52.3/88.7 | 98.6/100.0/100.0 | 74.8/89.4/97.5 | 70.0/86.4/97.4 | 35.2/59.7/91.4 |
| SemStamp (2023) | 97.7/98.4/99.5 | 88.1/92.8/97.8 | 92.4/95.2/98.1 | 76.0/83.0/94.5 | 92.5/95.8/98.3 | 80.9/87.9/95.9 | 77.9/86.3/95.7 | 69.2/80.0/92.4 |
| k-SemStamp (2024) | 99.0/99.0/99.7 | 45.5/64.1/92.4 | 53.9/70.3/94.8 | 43.5/61.9/91.6 | 100.0/100.0/99.9 | 49.2/68.4/93.2 | 54.8/71.6/93.7 | 43.9/61.2/89.2 |
| SimMark (2025) | 78.0/89.4/97.9 | 19.8/40.6/83.6 | 28.8/46.2/85.9 | 23.4/45.0/84.9 | 70.8/89.6/97.9 | 12.4/31.2/81.5 | 14.6/37.3/84.4 | 22.6/47.9/86.4 |
| Ours(Online) | 100.0/100.0/99.9 | 94.4/98.2/99.6 | 96.6/98.8/99.7 | 96.8/99.0/99.7 | 100.0/100.0/99.9 | 93.0/96.8/99.4 | 93.6/97.6/99.4 | 95.2/98.8/99.7 |
| Ours(Offline) | 99.4/100.0/99.9 | 91.4/97.4/99.3 | 94.6/97.8/99.6 | 94.4/98.6/99.6 | 99.7/99.8/99.9 | 88.0/95.8/99.1 | 91.3/95.8/99.3 | 92.0/95.2/99.3 |

**Obs. ❶ PMARK demonstrates superior semantic robustness across different backbones and benchmarks.** As shown in Table 1, PMARK exhibits strong robustness to paraphrase attacks. Specifically, the TP@FP1% of online PMARK remains above 93.0% across different paraphrasers, substantially outperforming the baselines. For Mistral-7B, the TP@FP1% on BOOKSUM and C4 after GPT paraphrasing reaches 96.8 and 95.2, exceeding the best SWM baseline SemStamp by 20.8% and 26.0% and the best TWM baseline SIR by 55.6% and 31.8%. The offline PMARK also shows strong robustness to paraphrase attacks, with TP@FP1% remaining above 90.8%, consistently ranking second to the online PMARK. A similar conclusion can be drawn from Figure 3, where both online and offline PMARK excel in TP@FP1% under back-translation attack. Despite a few competitive baseline results, such as those achieved by KGW when attacked by Pegasus, we argue that online PMark demonstrates near SOTA performance across a variety of paraphrasers, while offline PMark offers highly competitive robustness against paraphrase attacks.

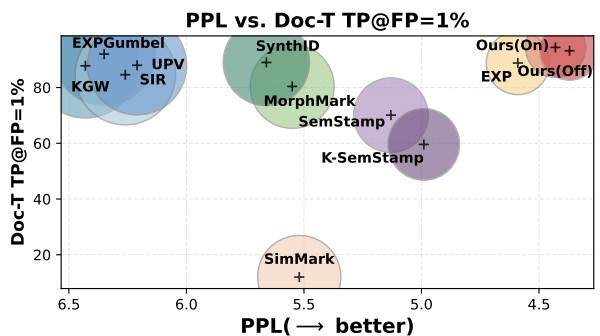

Figure 3: Results of Mistral-7B on the C4 dataset. Smaller bubbles denote lower PPL.

**Obs. ❷ PMARK achieves superior text generation quality.** As shown in Figure 3, the online version of PMARK attains the highest text quality on the BOOKSUM benchmark, with perplexity lower than the best baseline EXP (Aaronson, 2023) by nearly 0.7 on Mistral-7B. Among prior SWMs, k-SemStamp (Hou et al., 2024) achieves the lowest PPL (around 5.0) on BOOKSUM, yet it still underperforms PMARK by a clear margin. We observe similar trends on the C4 dataset: both the online and offline variants of PMARK deliver the highest-quality watermarked text. Similarly, EXP and k-SemStamp remain

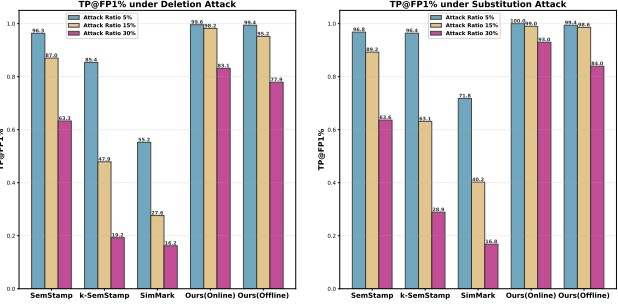

Figure 4: TP@FP1% under Word-D and Word-S attacks.

the strongest token-level and semantic-level baselines, respectively. Additional results on robustness and text quality are provided in Appendix H.1 and H.2.

**Obs. ❸ Previous SWM methods are vulnerable to word-level attacks.** We also evaluate PMARK under word-level deletion and synonym-substitution attacks with Mistral-7B on BOOK-SUM, as shown in Figure 4. Surprisingly, existing SWMs remain vulnerable to these token-level attacks because their watermark evidence is relatively sparse. As a result, even the reversal of one-bit watermark evidence can cause a large deviation in the $z$ statistic. In our work, we address this problem efficiently by applying multiple channel constraints to each sentence, rather than simply increasing generation length. Consequently, our method achieves state-of-the-art robustness: in online mode, TP@FP1% reaches 98.2% and 99.0% under 15% word deletion and synonym substitution, respectively; in offline mode, the corresponding TP@FP1% values are 95.2% and 98.6%.

## 5.3 COMPUTATION CONSUMPTION (RQ3)

**Obs. ❹ Evidence density is critical for SWM detectability.** Table 2 confirms our understanding of watermark evidence and sampling efficiency in SWM, evaluated with OPT-1.3B on BOOK-SUM. Specifically, online PMARK exhibits a high

Table 2: TP@FP1% across different sample budgets and channel numbers in online/offline modes.

| $N/b$ | b=1 | b=2 | b=3 | b=4 |
|---|---|---|---|---|
| $N = 8$ | 81.0/85.0 | 97.0/47.0 | 98.0/83.0 | -/- |
| $N = 16$ | 84.0/93.0 | 100.0/90.0 | 100.0/96.0 | 100.0/93.0 |
| $N = 32$ | 97.0/99.0 | 100.0/98.0 | 100.0/98.0 | 100.0/98.0 |
| $N = 64$ | 99.0/95.0 | 100.0/98.0 | 100.0/99.0 | 100.0/100.0 |

TP@FP% of 98% with an extremely limited sampling budget $N = 8$, supported by 3 channels. This suggests that dense watermark evidence is necessary for reducing sampling budget. We also note that offline PMARK requires a budget of 16 to achieve practical detectability, which we attribute to the high risk of prior deviation caused by a limited budget. While Bayesian models can improve accuracy in such probability estimation as shown in SynthID, we only present the naive results here and leave related improvements for future work.

**Obs. ❺ PMARK achieves the remarkable trade-off between token efficiency and performance among SWM methods.** To further verify the efficiency of

Table 3: Token efficiency of SWM methods. $T_s$ denotes average token consumption per sentence, while $T_t$ denotes consumption per token.

| Method | k-Sem | SimMark | Sem | Ours(On) | Ours(Off) |
|---|---|---|---|---|---|
| $T_s$ | 246.9 | 186.7 | 1694.4 | 315.8 | 239.7 |
| $T_t$ | 13.3 | 8.1 | 99.3 | 16.0 | 12.1 |

PMARK, we conducted additional experiments on token efficiency across existing SWM methods at the same detectability level, as shown in Table 3. While SimMark achieves the lowest token consumption during watermark generation, it performs poorly without well-tuned hyperparameters, as shown in Table 1. Similarly, k-SemStamp requires fewer tokens than online PMARK, but its robustness to paraphrasing is limited. In contrast, SemStamp exhibits the highest robustness to semantic attacks among existing SWMs, but at the cost of an impractical 1694.4 tokens per sentence. As a result, we argue that PMARK achieves the best trade-off among SWM methods: the offline mode requires fewer tokens than k-SemStamp while maintaining competitive robustness, and the online mode achieves the highest robustness with only 20% additional token consumption.

## 5.4 SENSITIVITY ANALYSIS (RQ4)

**Obs. ❻ Soft-$z$-test boosts the detection accuracy of online PMARK.** We present the parameter sensitivity analysis of two hyperparameters in Figure 5. Results of Doc-P(GPT) are also included for comparison. While the soft-$z$-test yields a stable improvement in robustness compared with the naive $z$-test ($K = +\infty, \delta = 0$), we find it is more beneficial for online PMARK, where increases of 11.9% and 6.4% in attacked TP@FP1% are ob-

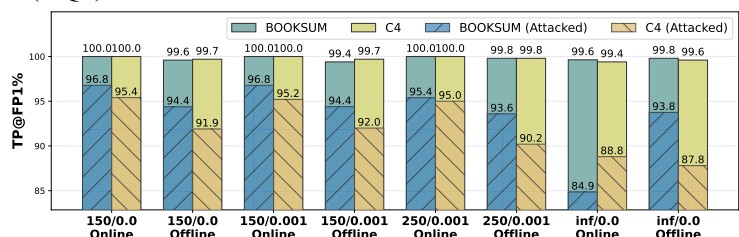

Figure 5: Performance of Mistral-7B under different hyperparameter settings ($K = 150, 250, +\infty$ and $\delta = 0, 0.001$).

served on BOOKSUM and C4, respectively. More results and cases can be found in Appendix H.4.

## 6 CONCLUSION

The semantics of the next sentence is an abstract and complex concept. To effectively estimate it indirectly, a proxy function that maps sentences from textual space to a scalar value is often used, either implicitly or explicitly. Building on this insight, we present a unified perspective on existing SWMs and propose a novel distortion-free paradigm based on median estimation. Furthermore, we identify that the vulnerability of current SWMs to adversarial attacks arises from the sparsity of watermark evidence. To address this issue, we introduce multi-channel constraints to enhance robustness. To the best of our knowledge, this is the first work to provide a unified theoretical framework for SWMs, the first distortion-free SWM approach, and the first to highlight and realize dense watermark evidence in SWM. Experimental results demonstrate that PMARK consistently outperforms existing SWM baselines in both text quality and robustness, offering a more reliable and effective paradigm for semantic-level watermarking.

## 7 ACKNOWLEDGMENTS

We would like to express our gratitude to the Red Bird MPhil Program at the Hong Kong University of Science and Technology (Guangzhou) for providing support, resources, and funding, which have been instrumental in the successful completion of this research.

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

## A  USE OF LLMS

We used AI assistants for two purposes: (1) generating routine code and boilerplate functions, which were subsequently reviewed and debugged by humans, and (2) performing grammatical review and sentence-level editing of the manuscript. The research methodology, findings, and analysis were independently proposed and conducted.

## B  DETAILED DEFINITION AND ADDITIONAL PROOFS

**Lemma 6** (Countable). *Let $\Sigma$ be a finite tokenizer vocabulary, and let $\Sigma^\star$ denote the set of all semantically complete sentences (including the null sentence) represented as finite token sequences over $\Sigma$. Then $\Sigma^\star$ is countable.*

*Proof.* Since $\Sigma$ is finite, fix a bijection $e : \Sigma \to \mathbb{N}_{\geq 1}$ (i.e. the index on the tokenizer). Let $(p_k)_{k \geq 1}$ be the increasing sequence of prime numbers. Define $C : \Sigma^{<\omega} \to \mathbb{N}$ by

$$C(s) := \begin{cases} 1, & s = \varnothing, \\ \displaystyle\prod_{k=1}^{|s|} p_k^{e(x_k)+1}, & s = (x_1, \ldots, x_{|s|}) \in \Sigma^{<\omega}. \end{cases} \tag{15}$$

If $C(s) = C(t)$, the Fundamental Theorem of Arithmetic forces the exponent of each $p_k$ to coincide on both sides. Hence $|s| = |t|$ and $e(x_k) = e(y_k)$ for all $k$, which implies $x_k = y_k$ by bijectivity of $e$, so $s = t$. Therefore $\Sigma^{<\omega}$ injects into $\mathbb{N}$ and is countable. Since $\Sigma^\star \subseteq \Sigma^{<\omega}$, it is countable as well. $\square$

**Definition 4** (Partition of a Set). *Let $U$ be a non-empty set. A partition $\mathcal{P}(U)$ of $U$ is a collection of non-empty, pairwise disjoint subsets of $U$ whose union equals $U$. Formally,*
$$\mathcal{P}(U) = \{A_1, \ldots, A_n\}$$
*such that:*

- *$A_i \neq \emptyset$ for all $i$,*
- *$A_i \cap A_j = \emptyset$ for all $i \neq j$,*
- *$\bigcup_{i=1}^n A_i = U$.*

*Each element $u \in U$ belongs to exactly one subset $A_i \in \mathcal{P}(U)$, which we denote as $\mathcal{P}_u(U)$ in the main body.*

**Proof of Lemma 1.**  Here we suppose that $q(S) > 0$ in practice, meaning that reject-sampling will always succeed when $N$ is large enough.

*Proof.* Let $\mathcal{F} : \Sigma^* \to U$ be the proxy with finite range $U$, and fix a green set $S \subseteq U$. For $u \in U$,

$$q(u) \triangleq \sum_{t \in \mathcal{F}^{-1}(u)} P_M(t \mid \boldsymbol{\pi}), \qquad q(S) \triangleq \sum_{\mathcal{F}(t) \in S} P_M(t \mid \boldsymbol{\pi}). \tag{16}$$

Hence

$$\Pr(Y = s \mid \boldsymbol{\pi}, S) = \sum_{k=0}^{\infty} \left(1 - q(S)\right)^k P_M(s \mid \boldsymbol{\pi}) = \frac{P_M(s \mid \boldsymbol{\pi})}{q(S)} \quad \text{for } \forall \mathcal{F}(s) \in S. \tag{17}$$

$\square$

**Proof of Theorem 2.**  Here the watermark code space is $E = \{S \mid S \subseteq U, |S| = \}$, where $|U| = M, = \gamma M$ and $|E| = \binom{M}{}$.

*Proof.* Based on equation 5 we have

$$P_M^w(s \mid \boldsymbol{\pi}) = \sum_{e_K \sim \mathrm{Un}(E)} P(Y = s \mid \boldsymbol{\pi}, e_K) P(e_K) = \frac{1}{\binom{M}{g}} \sum_{S \in E} P(Y = s \mid \boldsymbol{\pi}, S). \tag{18}$$

From Lemma 1,

$$P(Y = s \mid \boldsymbol{\pi}, S) = \begin{cases} \dfrac{P_M(s \mid \boldsymbol{\pi})}{q(S)}, & u \in S, \\ 0, & u \notin S, \end{cases} \tag{19}$$

therefore only sets $S$ containing $u = \mathcal{F}(s)$ contribute:

$$P_M^w(s \mid \boldsymbol{\pi}) = \frac{1}{\binom{M}{}} \sum_{\substack{S \subseteq E \\ u \in S}} \frac{P_M(s \mid \boldsymbol{\pi})}{q(S)} = P_M(s \mid \boldsymbol{\pi}) \cdot \frac{1}{\binom{M}{}} \sum_{\substack{S \subseteq U \\ |S|=, u \in S}} \frac{1}{q(S)}. \tag{20}$$

which is exactly equation 9. $\qquad\square$

**Proof of Corollary 2.1.** Denote equation 9 as

$$P_M^w(s \mid \boldsymbol{\pi}) = P_M(s \mid \boldsymbol{\pi}) A(u),$$

where

$$A(u) \triangleq \frac{1}{\binom{M}{}} \sum_{\substack{S \subseteq U \\ |S|=m, u \in S}} \frac{1}{\sum_{v \in S} q(v)}. \tag{$\star$}$$

So $P_M^w(\cdot \mid \boldsymbol{\pi})$ is distortion-free if and only if $A(u) \equiv 1$ for all $u \in U$.

*Proof.*
*Sufficiency.* If $q(u) \equiv 1/M$ for all $u$, then for every $S$, $q(S) = \sum_{v \in S} q(v) = g/M = \gamma$. Also $\Pr(u \in S) = \frac{\binom{M-1}{g-1}}{\binom{M}{}} = \frac{}{M} = \gamma$. Hence

$$A(u) = \frac{}{M} \cdot \frac{1}{\gamma} = 1,$$

which means

$$P_M^w(s \mid \boldsymbol{\pi}) = P_M(s \mid \boldsymbol{\pi}).$$

*Necessity.* Assume distortion-free, so $A(u) = 1$ for all $u$. Fix two indices $u \neq v$. Write the difference using ($\star$). Denote $T \subseteq U \setminus \{u, v\}$, $|T| = -1$, as a set containing both $u$ and $v$ will be neutralized in the difference, thus

$$A(u) - A(v) = \frac{1}{\binom{M}{}} \sum_{\substack{T \subseteq U \setminus \{u,v\} \\ |T|=-1}} \left( \frac{1}{q(u) + \sum_{w \in T} q(w)} - \frac{1}{q(v) + \sum_{w \in T} q(w)} \right).$$

Since $f(x) = \frac{1}{x+a}$ is a strictly decreasing function for any fixed $a > 0$, therefore

- If $q(u) > q(v)$, then every term in the sum is negative and hence $A(u) < A(v)$.
- If $q(u) < q(v)$, then $A(u) > A(v)$.

Distortion-free requires $A(u) \equiv A(v)$, hence we must have $q(u) = q(v)$. Since $u, v$ were arbitrary, all $q(\cdot)$ are equal, meaning that $q(u) = \frac{1}{M}$ for every $u \in U$. $\qquad\square$

**Proof of Theorem 4.** Let $\vec{x} = \frac{\mathcal{T}(s)}{\|\mathcal{T}(s)\|}$, $\vec{x}' = \frac{\mathcal{T}(\mathcal{A}(s))}{\|\mathcal{T}(\mathcal{A}(s))\|}$, and $\vec{v} = \frac{v}{\|v\|}$. Then the proxy is $f = \langle \vec{v}, \vec{x} \rangle \in [-1, 1]$ and $f' = \langle \vec{v}, \vec{x}' \rangle \in [-1, 1]$.

*Proof.* For unit vectors, $\|\vec{x}' - \vec{x}\|^2 = 2 - 2\langle \vec{x}', \vec{x} \rangle \leq 2d$, hence

$$|f' - f| = |\langle \vec{v}, \vec{x}' - \vec{x} \rangle| \leq \|\vec{v}\| \|\vec{x}' - \vec{x}\| \leq \sqrt{2d}.$$

A flip is possible only if the original score sits within the $\sqrt{2d}$ band around the median:

$$|f - m_v| \leq |f' - f| \leq \sqrt{2d} \implies f \in [m_v - \sqrt{2d}, m_v + \sqrt{2d}].$$

Therefore,

$$P_{\text{rm}} \leq \mu_M \left( s \in \Sigma^* \Big| \mathcal{F}_v(s) \in [m_v - \sqrt{2d}, m_v + \sqrt{2d}] \right).$$

$\qquad\square$

**Proof of Lemma 5.**

*Proof.* Without loss of generality, fix $x = e_1 = (1, 0, \ldots, 0)$. Draw $y$ uniformly on $S^{d-1}$. The angle $\theta$ between $x$ and $y$ satisfies $\cos\theta = x \cdot y = g(y)$. For $t \in (-1, 1)$, the set $\{y \in S^{d-1} : g(y) = t\}$ is a $(d-2)$-sphere of radius $\sqrt{1 - t^2}$. Its $(d-2)$-dimensional surface area is

$$\mathrm{A}(t) = \mathrm{A}(S^{d-2})(1 - t^2)^{\frac{d-3}{2}},$$

where $A(\cdot)$ denotes the surface area, $A(S^k) = \dfrac{2\pi^{(k+1)/2}}{\Gamma((k+1)/2)}$. Hence the probability that $T \in [t, t+dt]$ is proportional to that area times $dt$.

$$f_T(t) = \frac{A(S^{d-2})}{A(S^{d-1})} (1-t^2)^{\frac{d-3}{2}} = \frac{\Gamma\left(\frac{d}{2}\right)}{\Gamma\left(\frac{d-1}{2}\right)\sqrt{\pi}} (1-t^2)^{\frac{d-3}{2}}, \quad t \in [-1, 1].$$

With $t = \cos\theta$, $dt = -\sin\theta\, d\theta$, and $1 - t^2 = \sin^2\theta$. Therefore

$$p_d(\theta) = f_T(\cos\theta)\sin\theta = \frac{\Gamma\left(\frac{d}{2}\right)}{\Gamma\left(\frac{d-1}{2}\right)\sqrt{\pi}} \sin^{d-2}\theta, \quad \theta \in [0, \pi].$$

$\square$

**Theorem 7** (Multi-channel Robustness). *Let $\epsilon$ be an upper bound on $p = \mu_M(s \in \Sigma^* \mid |\mathcal{F}_{v_j}(s) - m_{v_j}| \leq \sqrt{2d})$ for all channels $j$, with proxy function independence across channels. Then:*

$$\mathbb{E}[z'] \geq (1-2\epsilon)\sqrt{bT}, \quad Var[z'] \leq 4\epsilon(1-\epsilon), \quad SNR \geq \frac{(1-2\epsilon)\sqrt{bT}}{2\sqrt{\epsilon(1-\epsilon)}}$$

*Proof.* Attack removes evidence in each channel with probability $\epsilon$:

$$\mathbb{E}[c'_{(t,j)}] = 1 - \epsilon, \quad Var[c'_{(t,j)}] = \epsilon(1-\epsilon)$$

Total evidence after attack:

$$\mathbb{E}[N'_g] = bT(1-\epsilon), \quad Var[N'_g] = bT\epsilon(1-\epsilon)$$

Let $X = N'_g - 0.5bT$:

$$\mathbb{E}[X] = bT(0.5 - \epsilon), \quad Var[X] = bT\epsilon(1-\epsilon)$$

By Jensen's inequality for convex $|\cdot|$:

$$\mathbb{E}[z'] = \mathbb{E}\left[\frac{|X|}{0.5\sqrt{bT}}\right] \geq \frac{|\mathbb{E}[X]|}{0.5\sqrt{bT}} = (1-2\epsilon)\sqrt{bT}$$

For variance bound, $X$ approximately $N(\mu_X, \sigma_X^2)$ with $\mu_X > 0$:

$$Var[|X|] \leq \sigma_X^2$$

$$Var[z'] = \frac{4}{bT}Var[|X|] \leq \frac{4}{bT} \cdot \sigma_X^2 = 4\epsilon(1-\epsilon)$$

Signal-to-noise ratio:

$$SNR = \frac{\mathbb{E}[z']}{\sqrt{Var[z']}} \geq \frac{(1-2\epsilon)\sqrt{bT}}{2\sqrt{\epsilon(1-\epsilon)}}$$

$\square$

This theorem demonstrates that increasing the number of channels $b$ improves robustness under the same attack strength. The signal-to-noise ratio grows with $\sqrt{bT}$, showing that multi-channel constraints enhance detectability. The approximation $Var[|X|] \leq \sigma_X^2$ is reasonable when $\mu_X/\sigma_X = \sqrt{bT} \cdot \frac{0.5-\epsilon}{\sqrt{\epsilon(1-\epsilon)}}$ is large, which occurs for typical experimental parameters ($b = 4, T \approx 10 - 20, \epsilon \leq 0.3$), yielding $\mu_X/\sigma_X \geq 2.8$. This theoretical analysis complements the empirical results showing PMARK's superior robustness against various attacks.

## C  ADDITIONAL RELATED WORK

Text watermarking techniques for large language models can be broadly categorized into two main paradigms: token-level methods and sentence-level methods.

### C.1  TOKEN-LEVEL TEXT WATERMARKING

Token-level watermarking, the more traditional approach, embeds signals by manipulating the probability distribution of tokens during the decoding process. A foundational method in this category partitions the vocabulary into "green" and "red" lists, boosting the probability of green-listed tokens to create a detectable statistical bias (Kirchenbauer et al., 2023). Numerous subsequent works have built upon this concept. For instance, some methods focus on enhancing security and text quality by adaptively watermarking only high-entropy tokens (Liu & Bu, 2024; Lu et al., 2024), while others dynamically adjust watermark strength to balance effectiveness and quality (Wang et al., 2025). To improve robustness against spoofing attacks, researchers have explored contrastive representation

learning to generate semantic-aware token lists (An et al., 2025). Other variations introduce signals in different domains, such as sinusoidal perturbations in the token probability vector to guard against model distillation (Zhao et al., 2023b), or frequency-based signals detectable via STFT (Xu et al., 2024b). There are also some token-level watermarking techniques (Giboulot & Furon, 2024; Bahri et al., 2024) that introduce a scorer to select chunk candidates with better detectability, thereby facilitating watermarks for black-box LLMs.

Multi-bit watermarking has also been a significant focus, with techniques employing error-correction codes (Qu et al., 2025), probability balancing (Wang et al., 2023), and majority-bit-based list construction (Xu et al., 2025) to embed more complex information. Furthermore, some approaches frame watermarking as a problem of hypothesis testing, optimizing detection accuracy by coupling output tokens with pseudo-random generators (Huang et al., 2023). Despite their diversity and ingenuity, a fundamental limitation shared by nearly all token-level methods is their vulnerability to semantic-level attacks. Since the watermark is tied to specific token choices, paraphrasing or other meaning-preserving modifications can easily disrupt or erase the embedded signal.

### C.2 Sentence-level Text Watermarking

To address the vulnerability of token-level methods to paraphrasing, sentence-level watermarking (SWM) has emerged as a more robust paradigm. These methods treat entire sentences as the fundamental unit for embedding information. A pioneering work in this area, SEMSTAMP (Hou et al., 2023), partitions the semantic embedding space using locality-sensitive hashing and employs rejection sampling to ensure that generated sentences fall into a pre-defined "valid" semantic region. This approach significantly enhances robustness against paraphrastic attacks.

Building on this idea, subsequent methods have explored various ways to define and select valid semantic regions. For example, SimMark (Dabiriaghdam & Wang, 2025) uses sentence embedding similarity, PersonaMark (Zhang et al., 2024) hashes sentence structures for personalization, and CoheMark (Zhang et al., 2025) leverages fuzzy c-means clustering and inter-sentence cohesion. Other approaches utilize advanced feature extractors such as sparse autoencoders (Yu et al., 2025) or fine-tuned LLM-based paraphrasers (Xu et al., 2024a) to embed multi-bit watermarks. However, a persistent challenge for most existing SWM techniques is their reliance on rejection sampling. Although effective for ensuring semantic validity, this mechanism often introduces significant distortion to the original text distribution, degrades text quality, and risks sampling failure when no valid candidates are found within a reasonable number of attempts. Furthermore, many of these methods lack rigorous theoretical guarantees for their robustness and distortion properties, a critical gap that our work aims to address.

**Concurrent Work.** A concurrent work is SAEMark (Yu et al., 2025), which introduces a concept analogous to the proxy function used in our framework, referred to as a "feature extractor." While both approaches share some conceptual similarities, PMARK offers notable advantages in several respects. (1) *Task setting.* SAEMark targets multi-bit watermarking that embeds additional information into the generated text, whereas PMARK is designed for single-bit watermarking. (2) *Text quality.* SAEMark selects sentences based on proximity to a target threshold, which may result in distortion. (3) *Robustness.* SAEMark embeds watermark signals on a per-sentence basis, which has been shown to be vulnerable to certain attacks in their original work, while PMARK leverages multi-channel constraints to significantly improve resistance to such threats. (4) *Flexibility.* PMARK allows for greater adaptability through customizable settings such as the pivot vector and the number of channels, which are more difficult to implement effectively within the SAEMark framework.

## D Additional Discussion

### D.1 Robust Random Seeds for Semantic-level Watermark

**Random Seeds in SWMs.** In previous SWM schemes such as SemStamp and k-SemStamp, the random seed for $s_n$ is extracted from the proxy value of the previous sentence $s_{n-1}$, as shown in Appendix E. However, this approach may not be truly random, since consecutive sentences often relate to the same topic (Nemecek et al., 2024). Moreover, SimMark (Dabiriaghdam & Wang, 2025) eliminates the use of a random seed entirely, relying instead on a fixed green region. These approaches can reduce the randomness of the key $k$ and thus break the assumption in 5. In this work, we use pre-defined random seeds that are de facto bound to specific positions $t$. This partially addresses the problem but leaves the challenge of achieving $n$-shot undetectability (Hu et al., 2023) unresolved.

**Definition 5** (Sentence-level $n$-shot-undetectable)**.** *For a fixed context $\boldsymbol{\pi}$, we say that a watermarked LLM distribution is $n$-shot-undetectable compared to the original LLM if, for all $s \in \Sigma^\star$,*

$$\sum_{i=1}^n P_M(s \mid \boldsymbol{\pi}) = \sum_{k \in K} P(K) \sum_{i=1}^n P_M^w(s \mid \boldsymbol{\pi}; k).$$

Imagine a user asking the model the same question $n$ times. An unwatermarked model or $n$-shot-undetectable scheme would generate answers $x_i, i = 1, \ldots, n$ i.i.d. from $P_M(s \mid \boldsymbol{\pi})$, while watermarking schemes such as SemStamp would always generate answers based on the same $k$, since the context is identical across queries. A stronger notion, $n$-*sequence distortion-free*, is proposed in SynthID, which removes the requirement of identical contexts in Definition 5.

**Definition 6** (Sentence-level $n$-sequence Distortion-free)**.** *For a sequence of $n$ prompts $\boldsymbol{\pi}_1, \ldots, \boldsymbol{\pi}_n$ and a sequence of $n$ responses $y_1, \ldots, y_n \in \Sigma^\star$, we say that a watermarked LLM distribution is $n$-sequence distortion-free if, for all $s \in \Sigma^\star$,*

$$\sum_{i=1}^n P_M(y_i \mid \boldsymbol{\pi}_i) = \sum_{k \in K} P(K) \sum_{i=1}^n P_M^w(y_i \mid \boldsymbol{\pi}_i; k; (\boldsymbol{\pi}_1, y_1), \ldots, (\boldsymbol{\pi}_{i-1}, y_{i-1})).$$

In token-level watermarking methods such as Unbiased (Hu et al., 2023) and SynthID (Dathathri et al., 2024), this is achieved by skipping previously seen contexts and directly outputting unwatermarked tokens.

However, in semantic-level watermark this is difficult to implement. Using the proxy of the previous sentence like SemStamp reintroduces the problem of reduced randomness in $K$. Furthermore, mixing SWM with token-level random seed generators may increase vulnerability to paraphrase attacks. Therefore, we consider semantic-level dynamic random seed generation to be a challenging open problem and leave it for future work. In this work, we directly adopt pre-defined random seeds following Christ et al. (2024) to achieve single-shot distortion-freeness.

## D.2 ROBUSTNESS TO CROSS-SENTENCE PARAPHRASING ATTACK

**Context-aware semantic-robust Random Seeds.** Based on the discussion in Appendix D.1, we further explore the potential of pseudo-random, context-aware, yet semantically robust random seeds within the PMARK framework. Specifically, we define $b$ additional orthogonal vectors $\zeta_1, \ldots, \zeta_b$, which are distinct from the pivot vectors $v_1, \ldots, v_b$ used during sampling.

The random seed $r_{(t,i)}$ for sentence $s_t$ is then based on the signal from the last $w$ sentence embeddings $\mathcal{T}(s_{t-w:t-1})$:

$$r_{(t,i)} = [\langle \zeta_i, \mathcal{T}(s_{t-w:t-1}) \rangle > 0] = \left[ \frac{\zeta_i \cdot \mathcal{T}(s_{t-w:t-1})}{\|\zeta_i\| \cdot \|\mathcal{T}(s_{t-w:t-1})\|} > 0 \right].$$

We leave the rigorous analysis of the distribution of these random seeds $r_{(t,i)}$ for future work. However, this approach is intuitively more random than seeds directly adapted from the proxy value of the last sentence, as employed by SemStamp and k-SemStamp.

Table 4: Results for baseline methods and PMARK on OPT-1.3B and Mistral-7B across the C4 and BOOKSUM benchmarks, under extreme paragraph-level paraphrasing attacks by GPT-3.5-turbo. For each dataset and model combination, we report TP@FP=1%, TP@FP=5%, and AUC from left to right. **Bold** denotes the best result, and underlined denotes the second-best.

| Method | (C4, OPT-1.3B) | (C4, Mistral-7B) | (BOOKSUM, OPT-1.3B) | (BOOKSUM, Mistral-7B) |
|---|---|---|---|---|
| KGW | 34.20/68.84/90.51 | 37.82/65.17/91.62 | 2.81/5.64/55.25 | 42.85/70.95/90.88 |
| UPV | 59.20/78.98/95.68 | 10.62/46.42/89.03 | 81.56/**88.66**/**97.86** | 35.41/60.83/90.51 |
| SynthID | 1.83/25.98/69.14 | 7.11/12.82/66.33 | 4.45/17.69/62.15 | 5.09/12.19/64.88 |
| KSEMSTAMP | 0.00/0.81/53.77 | 0.00/2.26/50.79 | 2.60/8.44/56.38 | 9.60/15.96/54.64 |
| **Ours(Online)** | 88.46/**96.15**/97.63 | **96.97/100.00/99.91** | **82.30/94.44**/97.61 | 75.00/87.50/98.18 |
| **Ours(Offline)** | **75.00**/95.83/**98.26** | 41.67/63.95/91.64 | 70.83/82.11/96.33 | **77.45/95.65/98.39** |

In Table 4, we present the results of PMARK under paragraph-level paraphrasing attacks with $w = 3$, an attack setting where most existing watermark methods are nearly powerless. The results demonstrate the potential of PMARK under extreme cross-sentence paraphrasing attacks, where 1) Online PMARK exhibits a remarkable TPR@FPR1% $> 75\%$ across different backbones and datasets; 2) Offline PMARK maintains a highly competitive performance compared with baselines.

**Experiment Details.** The prompt used to instruct gpt-3.5-turbo for paragraph-level paraphrasing is as follows:

```
<system>
You are a helpful assistant to rewrite the text.
</system>
<user>
Please rewrite the following text, avoiding the use of
same words or phrases as the original text as much as possible.
You are able to merge sentences or change their order:
{whole_text}
</user>

<Original Watermarked text>
The troupe's goal is to transform Passepartout into a clown-like
character. Passepartout's story starts in the medieval town of
Qeynan. He is being held captive by a man named Laidrin. He is a
sorcerer who makes Passepartout into a clown-like character.
Passepartout's story begins with a peasant, Fath, who has been
betrayed by the king's men. Fath flees with his wife ...
<Is_watermarked: True>

<Detected Text after Attack>
The troupe aims to transform Passepartout into a clown-like figure.
The story begins in the medieval town of Qeynan, where Passepartout
is held captive by Laidrin, a sorcerer who casts a spell on him,
turning him into a clownish character. Meanwhile, a peasant named
Fath, betrayed by the king's men, flees into the forest with ...
<Is_watermarked: True>
```

# E ANALYSIS OF EXISTING SWM METHODS

In Section 2, we presented a brief analysis of existing SWM methods, whose core lies in the proxy functions they employ.

**SemStamp.** Specifically, the proxy function of SemStamp can be defined as

$$\mathcal{F}_{\text{SemStamp}} = \text{LSH}(\mathcal{T}(s)) = \left[ \text{LSH}_1(\mathcal{T}(s)) \parallel \cdots \parallel \text{LSH}_n(\mathcal{T}(s)) \right],$$

where $\text{LSH}_i(v) = \text{sign}(u_i^\top v)$ (Indyk & Motwani, 1998; Charikar, 2002), $\{u_i\}_{i=1}^n \subset \mathbb{R}^d$ is a set of randomly initialized vectors, and $\mathcal{E}$ is a text encoder with $\mathcal{T}(s) \in \mathbb{R}^d$. The Green Region $G_t$ for each sentence position $t$ is then defined as

$$G_t = U'[: \gamma \cdot 2^d], \quad U' = \text{Shuffle}(U; r_t), \quad r_t = q \cdot \mathcal{F}(s_{t-1}),$$

where $\text{Shuffle}(U; r)$ denotes shuffling the ordered list $U$ with random seed $r$, and $q$ is a large prime.

**k-SemStamp.** For k-SemStamp (Hou et al., 2023),

$$\mathcal{F}_{\text{k-SemStamp}} = \underset{j}{\text{argmax}} \frac{u_j^\top \mathcal{T}(s)}{\|u_j\| \cdot \|\mathcal{T}(s)\|},$$

where $\{u_j\}$ are cluster centers obtained via k-means clustering on the entire dataset. Similar to SemStamp (Hou et al., 2023), the Green Region $G_t$ is given by

$$G_t = U'[: \gamma \cdot 2^d], \quad U' = \text{Shuffle}(U; r_t), \quad r_t = q \cdot \mathcal{F}(s_{t-1}).$$

**SimMark.** For SimMark (Dabiriaghdam & Wang, 2025),

$$\mathcal{F}_{\text{SimMark}}(s_t) = \frac{\mathcal{T}(s_{t-1})^\top \mathcal{T}(s_t)}{\|\mathcal{T}(s_{t-1})\| \cdot \|\mathcal{T}(s_t)\|},$$

where $s_{t-1}$ is the previous sentence. A fixed interval is then used as the Green Region. Under the cosine similarity used above, $G_t$ is always

$$G_t = [0.68, 0.76].$$

Such a pre-defined threshold carries a high risk that no valid candidate sentence has a proxy lying within the Green Region. Moreover, these carefully tuned hyperparameters may perform poorly in other settings, as shown in Table 1.

## F    OFFLINE ALGORITHM

To describe the generation and detection process of the offline version of PMARK more precisely, we illustrate it in pseudocode, as shown in Figure 6.

**Theorem 8** (Distortion Bound of Offline PMark). *Let $p_j = P_M(\mathcal{F}_{v_j}(s) > 0 \mid \pi)$, $|p_j - 0.5| \leq \epsilon$, $\delta_{TV} = \frac{1}{2} \sum_{s \in \Sigma^*} |P_M^w(s \mid \pi) - P_M(s \mid \pi)|$. Then $\delta_{TV} \leq \epsilon$ for single-channel case.*

*Proof.* For single-channel case:

$$P_M^w(s \mid \pi) = \begin{cases} P_M(s \mid \pi) \cdot \frac{1}{2p} & \text{if } \mathcal{F}_v(s) > 0 \\ P_M(s \mid \pi) \cdot \frac{1}{2(1-p)} & \text{if } \mathcal{F}_v(s) < 0 \end{cases}$$

Therefore,

$$\delta_{\text{TV}} = \frac{1}{2} \left[ \sum_{s:\mathcal{F}_v(s)>0} P_M(s \mid \pi) \left| \frac{1}{2p} - 1 \right| + \sum_{s:\mathcal{F}_v(s)<0} P_M(s \mid \pi) \left| \frac{1}{2(1-p)} - 1 \right| \right]$$

$$= \frac{1}{2} \left[ p \cdot \frac{|1-2p|}{2p} + (1-p) \cdot \frac{|1-2p|}{2(1-p)} \right] = \frac{1}{2} \left[ \frac{|1-2p|}{2} + \frac{|1-2p|}{2} \right] = \frac{|1-2p|}{2} \quad (21)$$

Since $|p - 0.5| \leq \epsilon$, $|1 - 2p| \leq 2\epsilon$, thus $\delta_{\text{TV}} \leq \epsilon$. $\square$

For the multi-channel offline PMark, the generation process selects the sentence with the highest watermark evidence score $\text{score}(s) = \sum_{j=1}^{b} \mathbf{1}_{\mathcal{F}_{v_j}(s)>0} \cdot r_j$. The distortion depends on the joint distribution of the proxy function values across channels.

Assuming that the proxy functions are orthogonal and that the signs $\mathbf{1}_{\mathcal{F}_{v_j}(s)>0}$ are independent across channels for a random sentence $s$, the score distribution can be analyzed. If $p_j = 0.5$ for all $j$, the selection is uniform, and distortion is zero. If $|p_j - 0.5| \leq \epsilon$, the distortion can be bounded by considering the probability that the score deviates from its expectation. Empirical results show that the perplexity of offline PMark is close to that of the unwatermarked text, indicating that $\epsilon$ is small in practice, and thus the distortion is negligible.

## G    IMPLEMENTATION DETAILS

### G.1    BASELINES

**MarkLLM.** We use the official implementation of MarkLLM (Pan et al., 2024)[1] to reproduce the results of various token-level baselines, including KGW (Kirchenbauer et al., 2023), UPV (Liu et al., 2023a), MorphMark (Wang et al., 2025), SIR (Liu et al., 2023b), EXP (Aaronson, 2023), EXPGumbel (Aaronson, 2023), and SynthID (Dathathri et al., 2024). During generation, we set `max_new_tokens` to 256, comparable to the generation length used by semantic-level methods. The temperature and `top_p` are fixed at 0.7 and 0.95, respectively, across all experiments to ensure fair comparison. In addition, for baselines that depend on external networks (e.g., SIR (Liu et al., 2023b) and UPV (Liu et al., 2023a)), we use the official weights provided by MarkLLM.

**k-SemStamp.** We adapt the official k-SemStamp implementation from MarkLLM to align it with other semantic-level methods. Specifically, we set `max_new_sentences` to 12 (instead of a token count), which is applied consistently across all semantic-level baselines. We set `max_trials` to the default value of 100 for SemStamp, k-SemStamp, and SimMark. For the embedding model, we use the fine-tuned `all-mpnet-base-v2` (Reimers & Gurevych, 2019)[2] provided by Hou et al. (2024). We also employ the k-means centroid weights[3] released by Hou et al. (2024), trained on the C4 (Raffel et al., 2020) and BookSum (Kryściński et al., 2021) datasets. For the backbone generation models OPT-1.3B (Zhang et al., 2022)[4] and Mistral-7B-v0.1 (Jiang et al., 2023)[5], we use the original Hugging Face versions to ensure fair comparison with all other methods. All other hyperparameters are kept at their default settings in the k-SemStamp implementation of MarkLLM. For all SWM experiments, we assume that sentence segmentation is consistent during both generation and detection, such that we evaluate only the intrinsic capability of the SWM method.

---

[1] https://github.com/THU-BPM/MarkLLM
[2] https://huggingface.co/AbeHou/SemStamp-c4-sbert
[3] https://github.com/abehou/SemStamp
[4] https://huggingface.co/facebook/opt-1.3b
[5] https://huggingface.co/mistralai/Mistral-7B-v0.1

**SemStamp.** For SemStamp (Hou et al., 2023), we use the authors' official implementation, setting temperature $= 0.7$, $\texttt{top\_p} = 0.95$, and $\texttt{max\_new\_sentences} = 12$ to match the other semantic-level methods. We also use the fine-tuned embedding model and the original generation model to ensure fair comparison.

**SimMark.** We use the official implementation of SimMark (Dabiriaghdam & Wang, 2025)[6]. We adopt cosine distance with the recommended validity interval [0.68, 0.76]. Additionally, we use the recommended embedding model $\texttt{instructor-large}$ (Su et al., 2023)[7] and the original generation models OPT-1.3B and Mistral-7B-v0.1 to maintain comparability with other methods. The factor $K$ is set to 250 for the soft-$z$ test, following the original paper.

**PMark.** During generation, we use the original $\texttt{all-mpnet-base-v2}$ embedding model (Reimers & Gurevych, 2019)[8] without fine-tuning. Pivot vectors are generated via QR decomposition of a Gaussian random matrix (Trefethen & Bau, 2022) to ensure orthogonality. We set temperature $= 0.7$, $\texttt{top\_p} = 0.95$, and $\texttt{max\_new\_sentences} = 12$, consistent with other semantic-level methods. The sample budget is $N = 64$, and the number of channels is $b = 4$. For detection, we set the numerical threshold $\delta = 0.001$ and the smoothing factor $K = 150$.

### G.2 Adversarial Attacks

**Doc-P.** We conduct paraphrase attacks following the official SemStamp implementation Hou et al. (2023; 2024). Specifically, for the paraphrasers Pegasus (Pu et al., 2023)[9] and Parrot (Sadasivan et al., 2023)[10], we adopt the bigram-style attack proposed by SemStamp Hou et al. (2023), which selects paraphrases with minimal bigram overlap relative to the source. For the GPT (OpenAI, 2022) paraphrasing attack, we use the prompt provided by MarkLLM (Pan et al., 2024). All attacks are applied at the sentence level across all methods and settings to simulate real-world scenarios.

**Doc-T.** We implement the back-translation attack following MarkLLM. Specifically, we use Llama-3.1-8B (Dubey et al., 2024)[11] to translate watermarked English sentences into Spanish and then back into English in an attempt to remove watermark evidence. However, as noted in Section 5.2, this attack can be biased and unstable in some cases.

**Word-D and Word-S.** We implement word deletion (Word-D) and synonym substitution (Word-S) attacks following the official MarkLLM implementation. We set the attack ratios to 5%, 15%, and 30% to evaluate the overall robustness of our method. Additional experimental results are reported in Appendix H.

## H Supplemented Experimental Results

### H.1 Robustness to Adversarial Attacks

We evaluate the robustness of existing watermarking methods under various attacks, with results shown in Table 5 and Table 6.

### H.2 Quality of Watermarked Text

Similar to Figure 3, we present additional results about the perplexity of existing methods across different benchmarks and backbones in Figure 7, 8 and 9.

### H.3 Sensitivity of Hyperparameter

We present additional hyperparameter analysis results for OPT-1.3B in Figure 10, where similar conclusions can be deduced.

### H.4 Case Study

We illustrate various cases of SemStamp and PMark across different backbones and datasets. Compared with the baseline, PMark demonstrates superior text quality and robustness, producing more diverse sentences and retaining watermark evidence even after paraphrase attacks.

---

[6] https://github.com/DabiriAghdam/SimMark
[7] https://huggingface.co/hkunlp/instructor-large
[8] https://huggingface.co/sentence-transformers/all-mpnet-base-v2
[9] https://huggingface.co/tuner007/pegasus_paraphrase
[10] https://github.com/PrithivirajDamodaran/Parrot_Paraphraser
[11] https://huggingface.co/meta-llama/Llama-3.1-8B

**Algorithm 3:** PMARK Offline Generation

**Input:** $M$; $s^{(0)}$; $\mathcal{T}$; $T$; $b$; $v^{(1)}, \ldots, v^{(b)}$; $R = \{r_{(t,j)}\}$; $N$
**Output:** $s^{(1)}, \ldots, s^{(T)}$
**for** $t \leftarrow 1$ **to** $T$ **do**
  **for** $j \leftarrow 1$ **to** $b$ **do**
    $\mathcal{F}_j(s) \leftarrow \langle v^{(j)}, \mathcal{T}(s) \rangle$
  **end**
  $u_{\max} \leftarrow -\infty$;   $B \leftarrow \varnothing$ **for** $i \leftarrow 1$ **to** $N$ **do**
    Sample $x^{(i)} \sim P_M(s \mid s^{(0:t-1)})$
    $\mathrm{Sig}(x^{(i)}) \leftarrow [\mathcal{F}_j(x^{(i)}) > 0]_{j=1}^{b}$
    $u^{(i)} \leftarrow \sum_{j=1}^{b} \mathrm{Sig}(x^{(i)})(j)$ **if** $u^{(i)} = b$ **then**
      $s^{(t)} \leftarrow x^{(i)}$
      **continue**
    **end**
    **if** $u^{(i)} > u_{\max}$ **then**
      $u_{\max} \leftarrow u^{(i)}$;   $B \leftarrow \{x^{(i)}\}$
    **end**
    **else if** $u^{(i)} = u_{\max}$ **then**
      $B \leftarrow B \cup \{x^{(i)}\}$
    **end**
  **end**
  Select $s^{(t)} \sim \mathrm{Un}(B)$
**end**
**return** $s^{(1)}, \ldots, s^{(T)}$

**Algorithm 4:** PMARK Offline Detection

**Input:** $S = [s^{(0)}, \ldots, s^{(T)}]$; $\mathcal{T}$; $v^{(1\ldots b)}$; $R = \{r_{(t,j)}\}$; $\delta$; $K$; $\alpha$
**Output:** True or False
**for** $j \leftarrow 1$ **to** $b$ **do**
  Define $\mathcal{F}_j(s) = \langle v^{(j)}, \mathcal{T}(s) \rangle$
**end**
$N_g \leftarrow 0$;   $N_{\text{total}} \leftarrow b \cdot T$
**for** $t \leftarrow 1$ **to** $T$ **do**
  **for** $j \leftarrow 1$ **to** $b$ **do**
    $x_{(t,j)} \leftarrow \mathcal{F}_j(s^{(t)})$;   $m_{(j)} \leftarrow 0$
    **if** $r_{(t,j)} = 1$ **and**
      $x_{(t,j)} > m_{(j)} - \delta$ **then**
      $c_{(t,j)} \leftarrow 1$
    **else if** $r_{(t,j)} = 0$ **and**
      $x_{(t,j)} < m_{(j)} + \delta$ **then**
      $c_{(t,j)} \leftarrow 1$
    **else**
      $c_{(t,j)} \leftarrow \exp\big(-K\,|x_{(t,j)} - m_{(j)}|\big)$
    **end**
    $N_g \leftarrow N_g + c_{(t,j)}$;
  **end**
**end**
$z \leftarrow \dfrac{|N_g - 0.5\,N_{\text{total}}|}{\sqrt{0.25\,N_{\text{total}}}}$; **return** $(z > z_\alpha)$

Figure 6: PMARK Offline Watermarking: Left: Generation using fixed prior median $= 0$; Right: Detection using soft z-test.

Table 5: Overall results for baseline methods and PMARK on OPT-1.3B and Mistral-7B on BOOK-SUM. **Doc-T** denotes back-translation attack by LLama-3.1-8B, while Word-D and Word-S are word-level attacks under different ratios.

| Method | No Attack↑ | Doc-P (Pegasus)↑ | Doc-P (Parrot)↑ | Doc-P (GPT)↑ | Doc-T↑ | Word-D(0.05)↑ | Word-D(0.15)↑ | Word-D(0.30)↑ | Word-S(0.05)↑ | Word-S(0.15)↑ | Word-S(0.30)↑ |
|---|---|---|---|---|---|---|---|---|---|---|---|
| **OPT-1.3B** | | | | | | | | | | | |
| EXP (2023) | 98.8/99.0/99.3 | 4.2/14.6/52.5 | 5.4/16.0/55.2 | 4.4/13.4/53.0 | 5.4/17.8/53.8 | 3.8/14.2/56.5 | 2.2/11.0/56.0 | 1.6/7.2/54.3 | 6.0/17.2/57.1 | 2.4/14.4/55.5 | 3.2/11.4/54.4 |
| EXPGumbel (2023) | 99.4/99.4/99.4 | 12.4/20.2/50.6 | 12.4/20.0/52.2 | 13.0/23.2/54.4 | 21.4/30.6/55.4 | 23.8/32.6/58.0 | 20.4/27.8/57.2 | 13.4/23.2/57.4 | 24.8/34.0/58.1 | 19.8/28.6/56.3 | 16.4/24.6/52.1 |
| KGW (2023) | 100.0/100.0/100.0 | 1.4/4.6/56.8 | 1.4/5.9/55.7 | 0.6/4.2/54.6 | 6.0/13.0/61.4 | 1.2/6.9/58.3 | 1.2/5.7/58.6 | 0.6/5.9/56.5 | 1.4/8.3/59.4 | 1.2/7.8/60.6 | 1.4/12.3/62.9 |
| SIR (2023b) | 99.6/100.0/99.8 | 84.2/93.8/97.5 | 80.6/93.8/98.1 | 49.8/82.8/95.9 | 77.4/89.8/95.6 | 96.2/98.8/99.6 | 86.0/93.4/96.9 | 65.2/78.6/89.5 | 98.6/100.0/99.8 | 92.2/98.4/99.5 | 69.8/88.6/95.7 |
| UPV (2023a) | 100.0/100.0/100.0 | 90.0/95.6/98.7 | 89.2/98.2/99.3 | 73.6/91.0/98.0 | 98.6/99.6/99.7 | 100.0/100.0/100.0 | 99.8/100.0/100.0 | 99.4/100.0/100.0 | 100.0/100.0/100.0 | 100.0/100.0/100.0 | 99.4/99.6/99.8 |
| SynthID (2024) | 100.0/100.0/99.9 | 38.2/56.6/86.9 | 25.8/43.6/83.8 | 1.8/14.6/65.7 | 96.0/97.0/99.2 | 99.2/100.0/99.9 | 95.4/98.0/99.5 | 56.0/77.4/94.5 | 99.6/100.0/99.9 | 95.6/98.8/99.6 | 54.8/76.0/94.5 |
| MorphMark (2025) | 100.0/100.0/100.0 | 1.8/7.7/55.2 | 1.2/6.1/54.6 | 0.8/4.2/51.1 | 5.0/13.3/60.0 | 1.8/7.3/57.9 | 1.2/7.2/56.7 | 1.2/7.5/57.0 | 2.0/9.2/58.7 | 3.4/10.5/60.0 | 1.0/8.2/61.6 |
| SemStamp (2023) | 97.7/98.8/99.4 | 89.0/93.2/97.3 | 90.7/93.0/97.5 | 79.6/85.4/94.2 | 87.8/92.8/97.7 | 97.6/98.2/99.3 | 89.4/94.4/98.4 | 67.5/80.8/95.1 | 97.7/98.6/99.5 | 93.2/96.8/99.1 | 78.1/86.6/96.4 |
| KSEMSTAMP (2024) | 99.6/100.0/99.9 | 74.5/84.6/97.4 | 72.3/86.6/97.5 | 62.1/72.9/95.2 | 72.3/84.2/97.2 | 93.8/97.0/99.6 | 64.7/76.8/96.3 | 29.1/44.3/86.7 | 99.2/99.8/99.9 | 81.8/90.6/98.5 | 44.1/59.3/92.5 |
| SimMark (2025) | 88.2/94.0/99.8 | 23.4/42.8/84.6 | 29.0/45.6/87.3 | 22.0/40.4/83.6 | 22.2/31.8/76.7 | 68.4/82.4/97.0 | 39.8/56.8/91.1 | 11.8/26.6/79.7 | 80.4/93.4/98.4 | 45.8/63.0/92.2 | 19.6/34.0/79.5 |
| Ours(Online) | 99.8/99.8/99.9 | 97.4/99.0/99.8 | 96.8/99.0/99.8 | 95.4/99.0/99.6 | 93.2/96.8/99.2 | 99.8/99.8/99.9 | 96.0/97.8/99.6 | 86.2/94.4/98.7 | 99.8/99.8/99.9 | 98.2/99.0/99.8 | 90.8/95.2/99.1 |
| Ours(Offline) | 99.4/99.6/99.8 | 93.2/98.2/99.5 | 95.2/98.6/99.5 | 94.2/98.8/99.6 | 96.4/98.8/99.6 | 99.8/99.8/99.9 | 98.0/99.0/99.7 | 86.7/96.0/99.0 | 99.4/99.6/99.9 | 98.4/99.4/99.8 | 87.0/96.6/99.4 |
| **Mistral-7B** | | | | | | | | | | | |
| EXP (2023) | 98.8/99.8/99.9 | 36.4/53.0/85.8 | 47.2/61.8/89.1 | 10.4/22.0/71.7 | 93.6/95.4/97.7 | 98.4/98.8/99.4 | 95.6/97.4/99.2 | 76.8/86.6/96.6 | 98.6/98.6/99.5 | 97.2/98.2/99.1 | 82.2/92.0/97.6 |
| EXPGumbel (2023) | 99.6/99.6/99.7 | 65.6/77.2/90.8 | 78.8/85.2/92.8 | 38.4/53.2/81.9 | 95.6/96.4/98.0 | 98.6/99.4/99.4 | 98.2/98.6/99.3 | 95.2/98.0/99.1 | 98.6/99.2/99.5 | 98.2/98.4/99.2 | 95.0/97.4/99.1 |
| KGW (2023) | 100.0/100.0/100.0 | 84.2/95.4/98.7 | 90.2/97.3/99.2 | 35.8/68.9/92.1 | 93.0/96.8/98.8 | 100.0/100.0/100.0 | 99.8/100.0/100.0 | 99.4/99.7/99.8 | 100.0/100.0/100.0 | 100.0/100.0/100.0 | 99.6/99.9/99.9 |
| SIR (2023b) | 100.0/100.0/99.9 | 85.0/92.6/98.0 | 80.8/92.0/98.2 | 41.2/71.4/93.6 | 81.2/91.0/96.9 | 97.8/99.6/99.8 | 90.0/95.6/99.0 | 72.4/81.2/93.1 | 99.6/100.0/99.9 | 92.0/97.8/99.4 | 73.0/87.0/96.5 |
| UPV (2023a) | 99.6/100.0/100.0 | 72.6/96.2/98.5 | 73.0/96.9/99.0 | 33.4/76.2/94.6 | 92.4/97.9/99.5 | 99.8/100.0/100.0 | 99.6/100.0/100.0 | 98.6/99.8/99.9 | 99.8/100.0/100.0 | 99.6/100.0/100.0 | 98.8/99.8/99.9 |
| SynthID (2024) | 100.0/100.0/100.0 | 30.2/49.2/83.5 | 28.2/48.4/85.5 | 3.0/13.0/65.0 | 95.8/97.4/99.4 | 99.6/99.8/99.9 | 96.4/98.6/99.7 | 63.4/80.6/95.2 | 99.6/99.8/99.9 | 97.4/98.4/99.6 | 53.2/74.4/94.5 |
| MorphMark (2025) | 99.8/100.0/100.0 | 68.2/88.9/97.2 | 77.4/93.3/98.4 | 24.6/52.8/87.7 | 89.0/93.5/97.4 | 100.0/100.0/100.0 | 99.4/99.9/99.9 | 93.8/99.0/99.6 | 100.0/100.0/100.0 | 99.0/100.0/99.9 | 92.2/98.7/99.6 |
| SemStamp (2023) | 97.7/98.4/99.5 | 88.1/92.8/97.8 | 92.4/95.2/98.1 | 76.0/83.0/94.5 | 87.1/94.2/98.5 | 96.3/98.0/99.4 | 87.9/94.2/98.6 | 63.4/76.8/93.9 | 96.7/98.4/99.3 | 89.4/95.0/98.7 | 64.4/77.6/94.6 |
| KSEMSTAMP (2024) | 99.0/99.0/99.7 | 45.5/64.1/92.4 | 53.9/70.3/94.8 | 43.5/61.9/91.6 | 67.3/79.8/95.9 | 85.4/92.8/98.9 | 47.9/64.5/93.1 | 19.2/33.1/80.0 | 96.4/97.8/99.5 | 63.1/80.0/96.2 | 28.9/47.3/88.2 |
| SimMark (2025) | 78.0/89.4/97.9 | 19.8/40.6/83.6 | 28.8/46.2/85.9 | 23.4/45.0/84.9 | 21.0/35.0/74.3 | 54.6/77.0/95.2 | 27.6/49.2/88.4 | 16.4/32.6/78.7 | 70.6/84.6/97.1 | 40.4/61.2/91.3 | 17.4/32.6/78.9 |
| Ours(Online) | 100.0/100.0/99.9 | 94.4/98.2/99.6 | 96.6/98.8/99.7 | 96.8/99.0/99.7 | 94.4/97.4/99.5 | 99.6/99.8/99.9 | 98.2/100.0/99.9 | 82.9/93.2/98.7 | 100.0/100.0/99.9 | 99.0/100.0/99.9 | 93.6/97.8/99.5 |
| Ours(Offline) | 99.4/100.0/99.9 | 91.4/97.4/99.3 | 94.6/97.8/99.6 | 94.4/98.6/99.6 | 95.8/99.2/99.7 | 99.4/100.0/99.9 | 95.4/99.4/99.8 | 78.9/95.8/98.6 | 99.4/99.8/99.9 | 98.6/99.6/99.8 | 84.9/95.2/98.3 |

## I BROADER IMPACTS AND LIMITATIONS

As a pioneering study on distortion-free and robust semantic-level watermarking, PMARK paves the way for more secure and powerful solutions for copyright protection and content attribution in the AI industry. However, several limitations remain. (1) For simplicity, we consider only fixed random seeds in this work, although more flexible techniques such as sliding windows and pseudo-random error-correcting codes could enhance adaptability. (2) While the sampling-based generation paradigm avoids the sampling failures of prior methods and provides an indirect estimate of the next-sentence distribution—thereby reducing distortion—the required sampling budget still needs to be further reduced for practical deployment. (3) Current SWM methods, including PMARK, operate at the sentence level, which depends on consistent sentence segmentation between the generation and

Table 6: Overall results for baseline methods and PMARK on OPT-1.3B and Mistral-7B on C4.

| Method | No Attack↑ | Doc-P (Pegasus)↑ | Doc-P (Parrot)↑ | Doc-P (GPT)↑ | Doc-T↑ | Word-D(0.05)↑ | Word-D(0.15)↑ | Word-D(0.30)↑ | Word-S(0.05)↑ | Word-S(0.15)↑ | Word-S(0.30)↑ |
|---|---|---|---|---|---|---|---|---|---|---|---|
| **OPT-1.3B** | | | | | | | | | | | |
| EXP (2023) | 99.0/99.6/99.8 | 54.6/68.2/86.9 | 42.4/59.6/84.6 | 21.2/37.2/73.3 | 90.6/93.8/96.9 | 98.0/98.2/99.5 | 96.0/97.8/99.2 | 72.0/85.6/95.8 | 98.4/98.8/99.6 | 96.8/98.2/99.5 | 80.4/89.4/96.9 |
| EXPGumbel (2023) | 98.6/98.6/99.3 | 75.2/84.6/91.8 | 74.6/81.0/92.8 | 55.6/66.0/86.0 | 94.2/95.6/97.4 | 97.2/98.0/99.0 | 97.2/97.8/98.8 | 95.0/95.8/98.1 | 97.8/98.2/99.1 | 96.8/97.4/98.7 | 94.0/95.4/98.2 |
| KGW (2023) | 100.0/100.0/100.0 | 89.3/97.0/99.2 | 76.2/91.4/98.1 | 51.4/78.5/95.0 | 90.5/94.6/97.8 | 100.0/100.0/100.0 | 100.0/100.0/100.0 | 99.5/100.0/100.0 | 100.0/100.0/100.0 | 100.0/100.0/100.0 | 98.5/99.8/99.9 |
| SIR (2023b) | 99.8/100.0/99.9 | 91.6/94.6/98.7 | 83.4/90.6/97.6 | 74.2/88.6/97.7 | 82.8/89.4/94.9 | 98.6/99.4/99.9 | 92.6/95.6/98.2 | 79.0/85.2/93.9 | 98.8/99.8/99.8 | 94.4/96.8/99.2 | 77.0/86.2/95.8 |
| UPV (2023a) | 100.0/100.0/100.0 | 90.6/95.4/98.3 | 78.7/88.4/97.4 | 80.9/91.3/98.2 | 98.2/98.8/99.5 | 100.0/100.0/100.0 | 100.0/100.0/100.0 | 99.4/99.8/99.9 | 100.0/100.0/100.0 | 100.0/100.0/100.0 | 99.9/100.0/99.9 |
| SynthID (2024) | 100.0/100.0/99.9 | 45.4/66.2/89.5 | 26.4/49.4/82.7 | 6.4/21.2/67.6 | 90.2/98.2/99.5 | 100.0/100.0/99.9 | 93.4/99.0/99.7 | 46.8/73.0/92.9 | 100.0/100.0/99.9 | 96.6/99.6/99.7 | 49.4/76.4/94.8 |
| MorphMark (2025) | 100.0/100.0/100.0 | 78.6/93.0/97.8 | 70.2/87.1/96.8 | 46.7/76.2/93.2 | 87.0/92.4/97.2 | 100.0/100.0/100.0 | 99.9/100.0/100.0 | 93.2/98.2/99.7 | 100.0/100.0/100.0 | 99.4/99.8/99.9 | 94.9/99.8/99.8 |
| SemStamp (2023) | 94.6/96.1/98.9 | 85.8/91.0/96.9 | 84.4/89.3/96.2 | 73.5/81.4/93.0 | 80.0/85.7/95.2 | 91.7/94.7/98.5 | 82.8/90.0/97.0 | 57.8/71.4/90.0 | 93.1/96.5/98.9 | 86.7/91.6/97.4 | 61.6/75.0/92.2 |
| KSEMSTAMP (2024) | 100.0/100.0/99.9 | 76.6/89.2/97.1 | 74.3/87.0/97.5 | 62.9/78.6/94.1 | 62.4/78.0/93.8 | 95.9/98.8/99.5 | 69.2/84.6/95.9 | 35.2/53.2/85.8 | 98.9/99.8/99.8 | 79.8/90.2/97.7 | 44.4/63.0/90.7 |
| SimMark (2025) | 77.6/94.2/98.5 | 11.6/35.8/82.9 | 13.6/39.2/84.5 | 16.0/44.8/86.5 | 11.0/26.6/75.3 | 52.2/83.4/96.6 | 27.4/57.0/90.7 | 8.8/27.4/81.4 | 66.6/88.2/97.5 | 30.0/60.0/91.4 | 9.8/28.2/77.7 |
| Ours(Online) | 100.0/100.0/99.9 | **96.2/99.6/99.8** | **97.2/99.0/99.8** | **97.8/99.6/99.8** | 95.2/98.0/99.5 | 100.0/100.0/99.9 | 97.2/98.4/99.7 | 85.6/94.2/98.7 | **100.0/100.0/99.9** | 98.4/99.6/99.9 | 87.6/96.0/99.2 |
| Ours(Offline) | 98.0/99.0/99.8 | 91.7/94.4/98.8 | 91.4/95.2/99.0 | 92.6/96.2/99.1 | 94.7/97.0/99.4 | 97.9/99.0/99.7 | 95.3/98.0/99.5 | 84.5/91.6/98.1 | 97.8/98.8/99.7 | 95.1/98.4/99.4 | 88.2/93.8/98.5 |
| **Mistral-7B** | | | | | | | | | | | |
| EXP (2023) | 99.2/99.2/99.5 | 44.4/60.4/85.7 | 34.2/54.8/84.2 | 17.4/29.8/72.1 | 88.8/92.6/97.6 | 98.2/98.8/99.4 | 93.2/96.0/98.7 | 63.0/75.6/94.2 | 98.2/99.0/99.4 | 95.6/98.6/99.3 | 74.0/85.2/96.5 |
| EXPGumbel (2023) | 98.6/98.6/99.3 | 71.2/82.0/92.5 | 66.2/77.2/90.8 | 40.2/55.4/82.5 | 92.0/93.8/97.2 | 98.2/98.4/99.0 | 97.2/98.6/99.0 | 91.4/95.8/98.3 | 98.2/98.2/99.0 | 92.3/98.2/99.0 | 92.6/96.4/98.4 |
| KGW (2023) | 100.0/100.0/100.0 | 85.6/95.5/98.8 | 79.2/93.7/98.8 | 54.2/76.7/95.6 | 87.8/92.9/96.5 | 100.0/100.0/100.0 | 100.0/100.0/100.0 | 99.4/99.9/99.9 | 100.0/100.0/100.0 | 99.8/100.0/100.0 | 99.0/100.0/99.9 |
| SIR (2023b) | 100.0/100.0/99.9 | 87.6/93.4/98.1 | 83.2/89.6/97.8 | 63.4/80.8/96.4 | 84.6/91.0/97.2 | 98.8/99.4/99.9 | 95.2/97.8/99.4 | 79.4/86.6/95.8 | 98.2/99.6/99.9 | 94.8/98.6/99.4 | 74.8/86.0/96.8 |
| UPV (2023a) | 99.4/99.8/99.9 | 71.4/91.4/97.7 | 61.9/85.4/97.5 | 34.9/68.6/93.4 | 88.0/95.6/98.9 | 100.0/100.0/100.0 | 99.2/100.0/99.9 | 97.0/99.8/99.9 | 99.8/100.0/100.0 | 99.4/100.0/99.9 | 95.6/99.8/99.8 |
| SynthID (2024) | 99.8/99.8/99.8 | 47.0/56.0/87.6 | 31.4/41.4/80.7 | 7.4/17.2/68.2 | 89.0/92.8/98.6 | 99.2/99.8/99.8 | 94.4/97.6/99.5 | 57.2/70.4/93.5 | 99.4/99.8/99.9 | 96.2/98.2/99.5 | 51.8/71.4/95.0 |
| MorphMark (2025) | 98.6/100.0/100.0 | 74.8/89.4/97.5 | 70.0/86.4/97.4 | 35.2/59.7/91.4 | 80.4/89.4/96.5 | 99.4/100.0/100.0 | 98.0/99.8/99.9 | 91.0/97.8/99.6 | 99.0/99.8/99.9 | 99.0/100.0/99.9 | 92.6/98.4/99.7 |
| SemStamp (2023) | 92.5/95.8/98.3 | 80.9/87.9/95.9 | 77.9/86.3/95.7 | 69.2/80.0/92.4 | 70.1/82.5/94.6 | 89.3/93.6/97.8 | 82.0/88.8/96.3 | 53.0/68.1/90.8 | 92.3/94.9/98.2 | 82.1/88.9/96.8 | 57.8/73.9/91.5 |
| KSEMSTAMP (2024) | 100.0/100.0/99.9 | 49.2/68.4/93.2 | 54.8/71.6/93.7 | 43.9/61.2/89.2 | 59.6/75.8/94.6 | 89.7/95.6/99.2 | 53.7/73.2/94.2 | 20.8/40.6/81.8 | 97.2/99.4/99.7 | 66.2/81.8/96.5 | 31.0/48.0/86.6 |
| SimMark (2025) | 70.8/89.6/97.9 | 12.4/31.2/81.5 | 14.6/37.3/84.4 | 22.6/47.9/86.4 | 12.0/22.9/67.9 | 50.8/76.8/95.8 | 27.0/51.9/89.2 | 13.8/31.6/80.7 | 64.6/85.3/97.2 | 28.6/54.7/90.5 | 10.8/25.6/74.1 |
| Ours(Online) | 100.0/100.0/99.9 | **93.0/96.8/99.4** | **93.6/97.6/99.4** | **95.2/98.8/99.7** | 93.6/97.2/99.2 | 100.0/100.0/99.9 | 98.2/99.4/99.7 | 85.8/93.0/98.6 | 100.0/100.0/99.9 | 98.6/100.0/99.9 | 89.4/96.4/99.4 |
| Ours(Offline) | 99.7/99.8/99.9 | 90.8/95.8/99.1 | 91.3/95.8/99.3 | 92.0/95.2/99.3 | **94.0/97.2/99.5** | 99.2/99.8/99.9 | 95.2/98.4/99.7 | 84.1/93.6/98.9 | 99.6/99.8/99.9 | 97.1/98.6/99.7 | 87.1/94.6/98.6 |

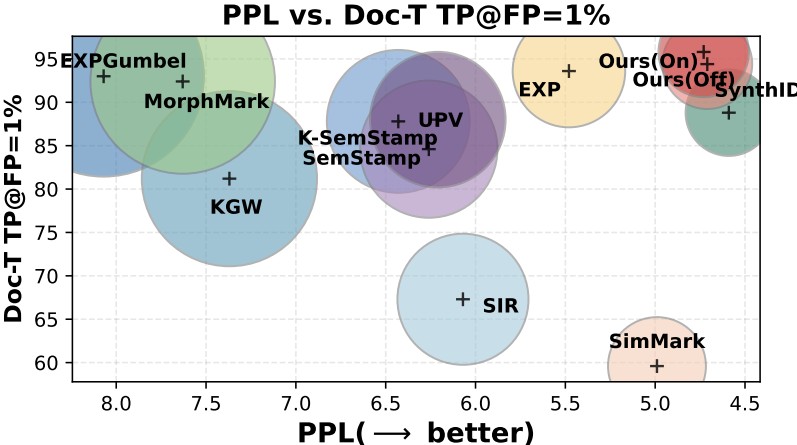

Figure 7: Mistral-7B PPL on BOOKSUM.

detection processes. Future research should explore more advanced SWM methods that eliminate this reliance and embed watermark evidence directly into the semantics of text, potentially enabling more sophisticated capabilities such as detecting whether specific ideas originate from LLMs.

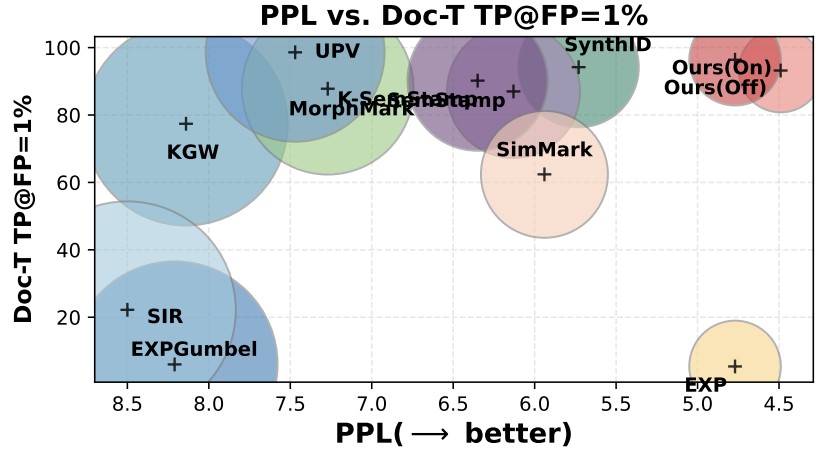

Figure 8: OPT-1.3B PPL on BOOKSUM.

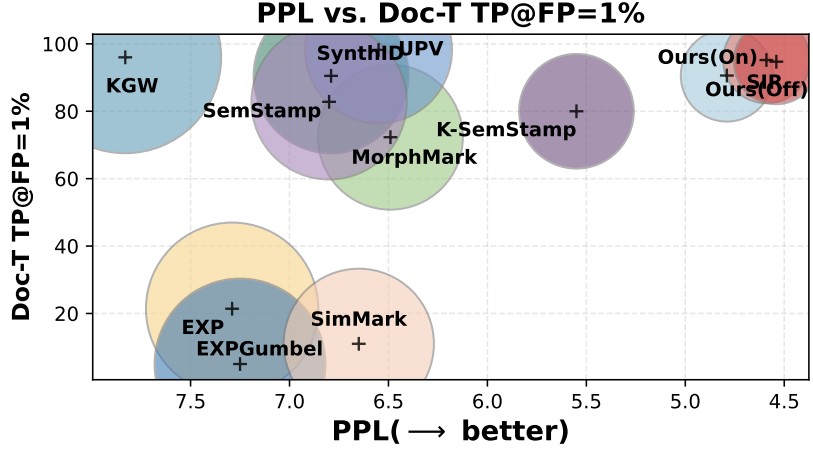

Figure 9: OPT-1.3B PPL on C4.

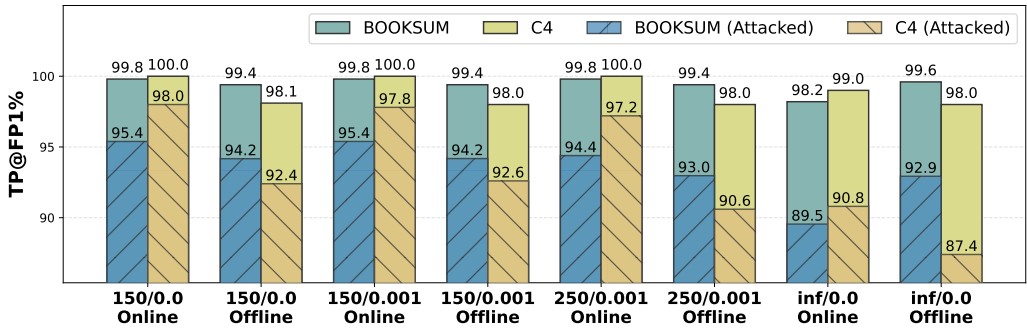

Figure 10: Performance comparison of OPT-1.3B under different hyperparameter settings.

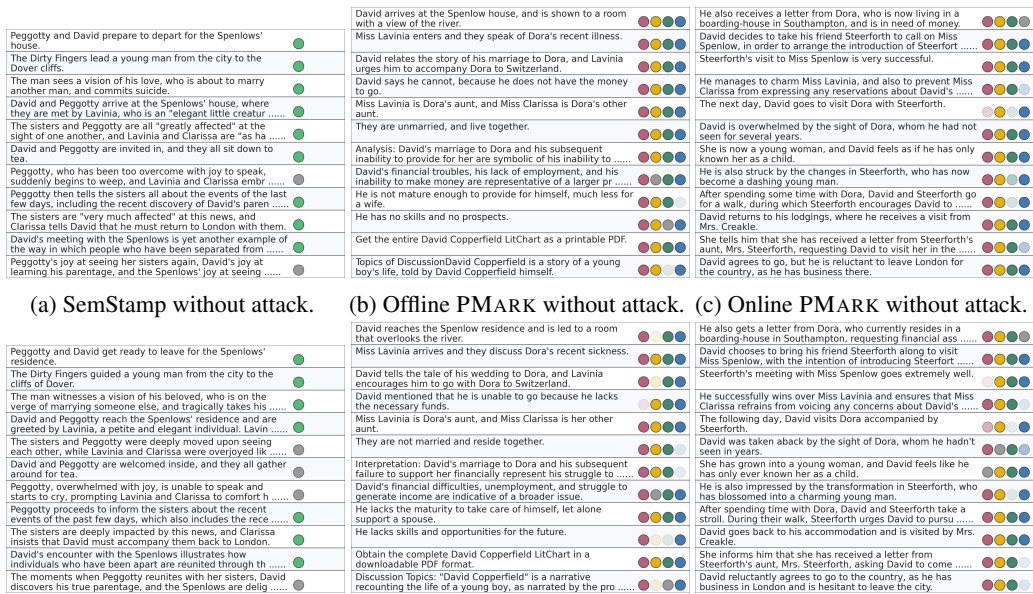

(a) SemStamp without attack. (b) Offline PMARK without attack. (c) Online PMARK without attack.

(d) SemStamp paraphrased by GPT. (e) Online PMARK paraphrased by GPT. (f) Offline PMARK paraphrased by GPT.

Figure 11: Cases from Mistral-7B on BOOKSUM dataset. ●, ●, ● and ● indicate valid watermark evidence is detected with the transpareteness denoting evidence strength, while ● indicates that no valid evidence detected in this channel.

(a) SemStamp without attack. (b) Offline PMARK without attack. (c) Online PMARK without attack.

(d) SemStamp paraphrased by GPT. (e) Online PMARK paraphrased by GPT. (f) Offline PMARK paraphrased by GPT.

Figure 12: Cases from Mistral-7B on C4 dataset.

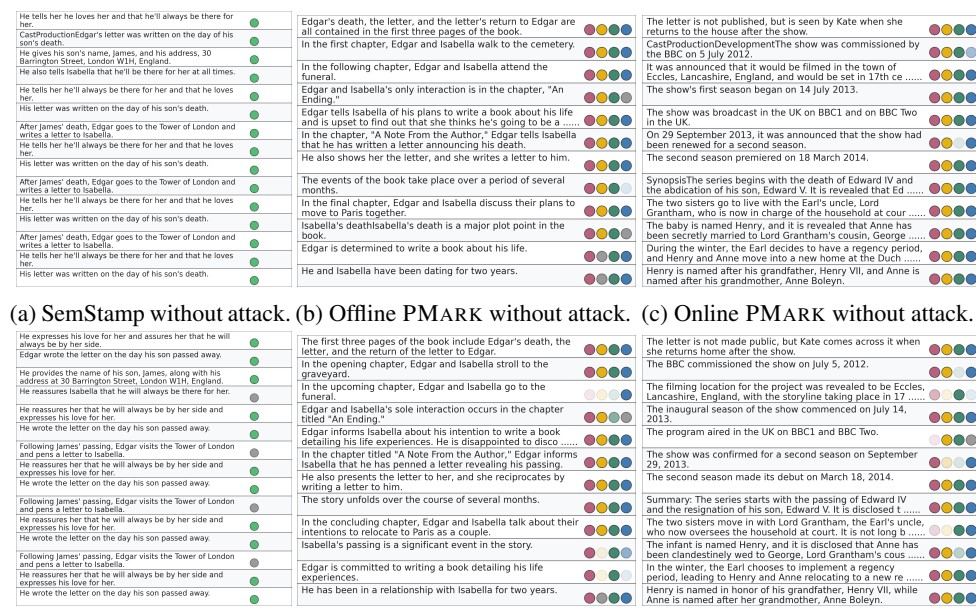

(a) SemStamp without attack. (b) Offline PMARK without attack. (c) Online PMARK without attack.

(d) SemStamp paraphrased by GPT. (e) Online PMARK paraphrased by GPT. (f) Offline PMARK paraphrased by GPT.

Figure 13: Cases from OPT-1.3B on BOOKSUM dataset.

(a) SemStamp without attack. (b) Offline PMARK without attack. (c) Online PMARK without attack.

(d) SemStamp paraphrased by GPT. (e) Online PMARK paraphrased by GPT. (f) Offline PMARK paraphrased by GPT.

Figure 14: Cases from OPT-1.3B on C4 dataset.

