# OpenReview forum: "PMark: Towards Robust and Distortion-free Semantic-level Watermarking with Channel Constraints"
_ICLR.cc/2026/Conference — ICLR 2026 Poster_

### Official Review · Reviewer_dwYt · 2025-10-31

**Soundness:** 3
**Presentation:** 3
**Contribution:** 3
**Rating:** 6
**Confidence:** 4

**Summary:**

PMARK is a semantic-level watermarking on a new Proxy Function (PF) framework: a PF maps sentences to scalars, enabling distortion-free sampling and detection. The method extends single-channel median sampling to a multi-channel scheme to improve the robustness against modification and paraphrasing.

**Strengths:**

1. The idea of using multi-channel to constrain the watermark to enhance the evidence is interesting.

2. The theoretical results seem solid.

3. Experimental results show superior performance over prior works.

**Weaknesses:**

1. Error bar is not presented for the main results. Some values are too close to those of prior works to claim the best.

2. Online and offline versions should be better considered as part of a general workflow and not require the user to make a decision.

3. Even though authors tried to solve the problem of watermarking short texts, which should be harder for this method, the requirements of computational resources, as well as time complexity, will be a problem.

**Questions:**

More analysis on the dependence of PF on the encoder.

More guidance on the hyperparameter settings from the theory to practical deployment.

---

> ### Author Response · Authors · 2025-11-19
> **Official Comment by Authors**
>
> We sincerely appreciate your positive recognition of our method’s solid theoretical foundations, comprehensive experiments, and overall design. We are also very grateful for your insightful feedback and provide detailed responses below.
>
> **Q1-1. Error bars are not presented for the main results.**
> **A1-1.** Thank you for pointing this out. Due to page limits, we omitted the error bars in the main text. We have now included the major standard deviations in the revised version for completeness.
> **Q1-2. Some values are too close to those of prior works to claim the best.**
> **A1-2.** We appreciate this observation. While PMark consistently achieves superior performance across multiple dimensions under paraphrase attacks, we acknowledge that some baselines may show competitive results in specific cases (particularly compared with the offline version). We have clarified these cases and adjusted our claims accordingly in the revised version. It is also worth noting that offline PMark still yields highly competitive results compared to baselines, with better generation quality and comparable latency.
>
> **Q2. The online and offline versions should be considered as part of a general workflow, not as separate options requiring user choice.**
> **A2.** Thank you for this valuable suggestion. Indeed, integrating the online and offline modes into a unified workflow is an interesting direction. In our current design, the online version offers higher detectability and robustness, while the offline version provides better efficiency. Practically, we recommend applying offline PMark to long-form generation tasks (e.g., reports or articles) and online PMark to short, high-precision scenarios (e.g., poems or short answers). We will explore automated switching and hybrid strategies between the two modes in future work.
>
> **Q3. The computational requirements and time complexity may pose challenges.**
> **A3.** Thank you for raising this concern. As discussed in Section 5.3, PMark achieves high computational efficiency among existing SWM methods, despite SimMark having lower overhead but very poor detectability and robustness. While sampling-based generation is commonly used in SWM due to the intractability of the sentence space $\Sigma^*$, we note that smaller models can serve as sampling proxies for larger ones when sampling (similar to speculative decoding), which will be a possible direction to explore in the future. We also foresee that some upcoming generation frameworks such as LCM (arXiv:2412.08821) and dLLM (arXiv:2502.09992) may provide more efficient solutions in SWM, which generate sentences or chunks as basic units.
>
> **Q4. More analysis on the dependence of the PF on the encoder.**
> **A4.** We agree that the choice of encoder is important. In PMark, we use cosine similarity as the proxy function since it naturally aligns with the encoder’s embedding space. Compared with SAEMark’s s SAE-based scorer(https://arxiv.org/abs/2508.08211), this design is simpler and more flexible for multi-channel extension. Our main experiments use all-mpnet-base-v2 without fine-tuning for fair comparison, and our preliminary single-channel tests with other encoders such as SONAR(https://arxiv.org/abs/2308.11466) and fine-tuned sbert(AbeHou/SemStamp-c4-sbert) show consistent trends.
> Additionally, as discussed in Lemma 5, PF values in high-dimensional spaces tend to concentrate near zero. We believe that shaping PF distributions to be more distinguishable (e.g., via PCA) may further improve detectability, which we leave for future exploration.
>
> **Q5. More guidance on hyperparameter settings from theory to practical deployment.**
> **A5.** Thank you for the question. We find that a standard configuration of N=64 (sample budget) and b=4 (channels) provides a strong balance between robustness and efficiency across datasets and models. Our hyperparameter sensitivity study also shows that results are stable across a wide range of settings. We recommend using the soft-z-test for robust detection, as it significantly improves performance, while the specific values of K and \delta have limited effect. Based on your suggestion, we will include detailed practical guidance for parameter config in our public code release afterwards to support community adoption.

---

### Official Review · Reviewer_SZC6 · 2025-11-01

**Soundness:** 3
**Presentation:** 3
**Contribution:** 3
**Rating:** 8
**Confidence:** 4

**Summary:**

This paper presents a unified theoretical framework for semantic-level watermarking (SWM) in natural text generation and introduces **PMARK**, a well-founded and practical method that is provably distortion-free and robust. As research on SWM remains nascent, partly due to limited theoretical guidance [1], this work could help pave the way for subsequent advances, even though several limitations warrant further study.

[1] Zhao, et al. Sok: Watermarking for ai-generated content. 2025 IEEE S&P.

**Strengths:**

1. Clean and solid theoretical analysis. The paper unifies prior work such as SemStamp, k-SemStamp, and SimMark under a proxy-function framework and presents a **provably distortion-free multi-channel** sampling rule furthermore.

2. Novel paradigm. This is the first distortion-free SWM method, whose robustness is enhanced by multi-channel constraints. The ideas in this work is elegant and effective, and may help deepen our understanding of semantic-level text watermarking.

3. Solid experiments. The experiments are solid, confirming the generation quality and robustness of PMARK. The ablation study is detailed and convincing.

4. Good writing and presentation. The paper is well organized and easy to follow; the figures and charts are clear, aside from a few minor issues (see Questions).

**Weaknesses:**

1. Theoretical analysis of multi-channel robustness. Although Eq. 11 characterizes the robustness of single-sentence watermarking, could the authors provide a more detailed theoretical analysis of **multi-channel** sampling under common conditions?

2. Discussion of token-level related work. Some token-level watermarks [2,3] use scoring functions during sampling as well. Including these works in the discussion would help contextualize the relationship to SWM.

3. There are also some limitations discussed in the paper including the analysis of offline method and the extension to n-shot undetectable SWM, which are leaved for future works by authors. Could the authors denote some possible directions for subsequent improvement?


[2] Giboulot, E. and Furon, T., 2024. WaterMax: breaking the LLM watermark detectability-robustness-quality trade-off. NeurIPS 2024.

[3] Bahri, D. and Wieting, J., 2024. A watermark for black-box language models. arXiv preprint arXiv:2410.02099.

**Questions:**

1. Although the authors claim that random seed generation for SWM is challenging and that prior approaches may not be suitable, could the authors suggest a possible solution or research direction for this problem?

2. As noted in the Weaknesses, could the authors provide a theoretical analysis of the adversarial robustness of multi-channel sampling?

3. (Minor) The annotations in Figure 4 are relatively small.

---

> ### Author Response · Authors · 2025-11-19
> **Official Comment by Authors**
>
> We sincerely appreciate your recognition of our solid theoretical analysis, well-designed method, comprehensive experiments, and clear presentation. We are grateful for your constructive feedback and provide our detailed responses below.
>
> **Q1. Theoretical analysis of multi-channel robustness.**
> **A1.** Thank you for the suggestion. In the revised version, we have added a new analysis as **Theorem 7** in **Appendix B** discussing the multi-channel setting under common assumptions. The results show that increasing the number of channels effectively enhances the signal-to-noise ratio during detection, further supporting the theoretical soundness of our multi-channel framework.
>
> **Q2. Discussion of token-level related work.**
> **A2.** We appreciate this valuable comment. In the revised version, we have expanded our related work section (**Appendix C.1**) to include token-level watermarking studies that also employ scoring functions during sampling, helping to better contextualize the connection between token-level and semantic-level watermarking.
>
> **Q3. Possible directions for subsequent improvement.**
> **A3.** Thank you for this insightful question.
> - For the analysis of offline method, as also mentioned by another reviewer, we have added a theoretical discussion in **Appendix F** analyzing the impact of the prior-median assumption theoretically.
> - Regarding n-shot undetectable SWM, we note that adopting a context-history mechanism similar to SynthID and Unbiased, together with a context-aware private key design (see A4), could address this problem effectively.
>
> **Q4. Possible solutions or research directions for random seed generation in SWM.**
> **A4.** We agree that random seed generation remains an open and important question. We propose two possible directions and provide some exploratory results additionally:
> - **Pseudorandom error-correcting codes:** Introducing structured redundancy can help recover dislocated seeds under cross-sentence paraphrasing. Our framework is well compatible with such a design.
> - **Context-aware semantic-robust keys:** As discussed in Appendix D, conventional hash-based seeders are sensitive to paraphrasing. We instead suggest using sentence-embedding–based, context-aware seeders to improve robustness against both paraphrasing and seed dislocation. Some exploring results have been attached in **Appendix D.2**, verifying the generalizability of our framework.
>
> **Q5. Theoretical analysis of adversarial robustness of multi-channel sampling.**
> **A5.** Please refer to Response 1 for the newly added theoretical discussion in **Appendix B**, which now explicitly covers multi-channel robustness under adversarial settings.
>
> **Q6. (Minor) The annotations in Figure 4 are relatively small.**
> **A6.** Thank you a lot for the observation. We plan to enlarge the annotations in Figure 4 in the next version to improve readability and presentation quality.

---

### Official Review · Reviewer_4inb · 2025-11-01

**Soundness:** 3
**Presentation:** 3
**Contribution:** 3
**Rating:** 6
**Confidence:** 4

**Summary:**

The paper introduces PMARK, a semantic-level watermarking method that aims to be both distortion-free and robust to paraphrasing. The core abstraction is a proxy function that maps a sentence to a scalar. PMARK employs rejection sampling on median-based half-partitioning of semantic space along several orthogonal channels. The authors prove distortion-freeness and demonstrate improved semantic robustness across different benchmarks.

**Strengths:**

1. The proxy function formalization and closed‑form distortion analysis clarify why split‑and‑reject SWMs distort and motivate a median-split alternative; the multi‑channel constraint is a simple, elegant idea that increases evidence density.
2. Strong results across two backbones, two datasets, many attacks, and against several prior methods; PMARK‑Online exceeds prior SWM baselines by large margins.

**Weaknesses:**

1. This work assumes the private key is drawn from a prior distribution. If the prior key is hashed from the context, the semantic robustness would break since the random seeds can't be recovered.
2. Multi-channel sampling introduces additional computation overhead and seems inherently incompatible with batch decoding.

**Questions:**

1. Can you provide an end‑to‑end evaluation where the detector does not query the generator (unknown M and prompts), using the offline variant?
2. Is there formal guarantees for the offline method?
3. How does PMARK work when an attacker obfuscates sentence segmentation, such as randomly changing all periods to commas?

---

> ### Author Response · Authors · 2025-11-19
> **Official Comment by Authors**
>
> We sincerely appreciate your recognition of our work, including the clear analysis, elegant design, and comprehensive experiments. We have carefully considered your valuable questions and provide detailed responses below.
>
> **Q1. This work assumes the private key is drawn from a prior distribution. If the prior key is hashed from the context, the semantic robustness would break since the random seeds can’t be recovered.**
> **A1.** We deeply appreciate this insightful observation regarding watermark key generation. As discussed in Appendix D, traditional context-hash-based prior keys, commonly used in token-level watermarking, are indeed vulnerable to token modifications. Additionally, deriving prior keys from the proxy value of the previous sentence like k-SemStamp reduces randomness (thus degrading generation quality greatly) and violates the assumptions required for theoretical guarantees.
> Although we leave this as an open problem, we'd like to provide two possible direction here:
> - **Pseudorandom error-correcting codes:** These can help recover dislocated seeds under cross-sentence paraphrasing by introducing error-correcting mechanism.
> - **Context-aware semantic-robust keys:** Context-aware seeding that leverages sentence embeddings rather than token hashes can serve as a robust alternative, compatible well with our framework. We have conducted explorary experiments on this idea and added the results to **Appendix D.2**, where PMark maintains high detectability (> 75%) much better than baselines (<50% commonly).
>
> **Q2. Multi-channel sampling introduces additional computation overhead and seems incompatible with batch decoding.**
> **A2.** In this work we acknowledge that existing SWM methods inherently require next-sentence sampling, as the true distribution over the sentence space $\Sigma^*$ is unknown. While batch decoding for SWM is an open research challenge, we provide two possible directions to mitigate the computation cost:
> - **Speculative decoding:** Since LLMs of different sizes but similar architectures share comparable semantic distributions, a smaller model can be used to sample candidate watermarked sentences to guide decoding in larger models (similar to speculative decoding approaches), thus reducing overhead.
> - **N-gram-level generation:** Our framework can also operate on n-gram chunks, preserving partial batch decoding advantages, as mentioned by Reviewer zq8u. Although ensuring consistent segmentation between generation and detection remains challenging under paraphrasing attacks.
>
> We also assume that future generation paradigms such as LCM (arXiv:2412.08821) and dLLM (arXiv:2502.09992) may provide more efficient solutions in SWM. Despite this, PMark is still one of the most efficient and powerful SWM methods, as shown in Section 5.3.
>
> **Q3. Can you provide an end-to-end evaluation where the detector does not query the generator (unknown M and prompts), using the offline variant?**
> **A3.** Yes. Actually in offline mode, the detector operates entirely without access to the generator, as described in Appendix F (Figure 6). We have now highlighted this feature more clearly in the main body to avoid confusion. Thank you for bringing this to our attention.
>
> **Q4. Is there any formal guarantee for the offline method?**
> **A4.** In this work, we mainly validate the offline PMark through empirical results. In response to your concern, we have added additional theoretical discussion in **Appendix F**, detailing the effect of the prior-median assumption and its approximation bounds. We hope these clarifications strengthen understanding of the offline method’s theoretical foundation.
>
> **Q5. How does PMARK perform when an attacker obfuscates sentence segmentation (e.g., replacing all periods with commas)?**
> **A5.** This is an interesting and less-explored question in semantic watermarking. Prior paraphrasers (e.g., Pegasus and Parrot) mainly perform sentence-by-sentence paraphrasing. To further evaluate robustness, we conducted additional experiments using GPT-based cross-sentence paraphrasing that includes sentence merging and reordering, under the dynamic key idea described in A1. The results in **Appendix D.2** confirm that our framework maintains good detection accuracy and excellent adaptability even when sentence segmentation is obfuscated, compared with baselines powerless for this setting.
> Regarding extreme artificial punctuation replacement (e.g., converting all periods to commas), such attack is somehow impractical in real-world scenarios for watermark tampering, and may be mitigated by using advanced neural segmentors that identify semantically complete units rather than relying solely on punctuation (i.e., segmenting a compound sentence into some simple sentences). In fact, developing such semantic-aware annotators is an exciting topic to further enhance generalizability of SWM methods.

---

### Official Review · Reviewer_zq8u · 2025-11-02

**Soundness:** 4
**Presentation:** 4
**Contribution:** 4
**Rating:** 8
**Confidence:** 4

**Summary:**

The paper first introduces a new theoretical framework on semantic-level watermarking with the concept of proxy functions, which covers existing methods.
It points out that they are not robust to attacks due to distribution distortions.
Building on the framework, they propose PMark, which is distortion-free.
The experiments show that PMark improves the robustness against paraphrasing-style attacks and also outperforms in text quality.

**Strengths:**

* The paper's theoretical framework is accurate and clear. It is very important for the community to systematically understand the literature and the limitations of the current stage.

* The experiments are comprehensive, and the settings are timely and accurate.

* The presentation of the paper is clear and fluent.

* From my knowledge, this is the first distortion-free semantic watermark paper.

**Weaknesses:**

* (a minor point) Some symbols look overlapped but with different meanings. I am unsure if there is any way to prevent it. For example, $m$ denotes the size of the green list in Theorem 2; $m_v$ denotes the median of $F$ later in L213. There might be a chance the readers build a connection between, but indeed they should not.

**Questions:**

* (It is an open question and not related to the assessment.) Are there any insights on how the method would work (or not) if we do cross-sentence paraphrasing attacks (e.g., there will be sentence deletion, merging, etc).

-- Will the smooth counting mechanism and the soft-count-based z-test help?

-- Why do you focus on the sentence-level instead of the n-gram chunk? The former may have a large uncertainty in each sentence's length. Maybe if a sentence only contains 2 words but has 1 word removed in attack, the sentence-level semantics will shift a lot.

---

> ### Author Response · Authors · 2025-11-19
> **Official Comment by Authors**
>
> We sincerely appreciate your positive feedback on our theoretical analysis, comprehensive experiments, clear presentation, and contribution. We highly value your review and have carefully discussed your comments to provide thoughtful and in-depth responses.
> **Q1. Some symbols look overlapped but with different meanings.**
> **A1.** Thank you for pointing this out. To improve clarity, we have replaced the symbol m (used for the size of the green list in Theorem 2 and Appendix B) with g, avoiding any potential confusion with m elsewhere in the paper.
>
> **Q2. Are there any insights on how the method would work (or not) under cross-sentence paraphrasing attacks (e.g., involving sentence deletion, merging, etc.)?**
> **Q2-1. Will the smooth counting mechanism and the soft-count-based z-test help?**
> **A2-1.** These mechanisms can help to a certain extent. Based on our experiments, the soft-counting mechanism indeed improves detection accuracy for semantic watermarking. However, it alone cannot fully address the core challenge of cross-sentence paraphrasing, namely, the dislocation of random seeds. In such cases, soft counting cannot by itself enhance robustness.
> Additionally, when combined with appropriately designed random seeds (e.g., redundant error-correcting codes or context-aware random seeders), the soft-counting mechanism may further improve resilience by extracting more watermark signals from attacked sentences.
>
> **Q2-2. Why do you focus on the sentence level instead of the n-gram level?**
> **A2-2.** Our choice is motivated by two main considerations:
> - **Methodological consistency:** During this work, we follow the paradigm established by prior semantic-level watermarking methods such as SemStamp, k-SemStamp, and SimMark, which all operate at the sentence level.
> - **Detection stability:** N-gram-level watermarking faces segmentation challenges during detection. For example, segmenting paraphrased text into consistent n-gram chunks is error-prone during detection.
> Although sentence-level watermarking is not flawless, sentence structure tends to remain more stable even under cross- or intra-sentence paraphrasing. This structural stability likely explains why most pioneering SWM methods also adopt the sentence as their basic unit.
> However, n-gram-level watermark is a very interesting idea that may help enhance the compatiblity of SWM methods with batch decoding (concern from reviewer 4inb), which could be a promising direction to explore in the future.
>
> **A2.** Further discussion on adapting PMark to cross-sentence paraphrasing attacks.
> We have conducted additional discussions to explore possible extensions of PMark to better handle such attacks. Our main ideas are as follows:
> - **Pseudorandom error-correcting codes:** Cross-sentence paraphrasing mainly disrupts random seed alignment. Properly designed error-correcting codes could enable detection even when seed dislocation occurs. The detailed design of such codes is beyond the scope of this paper but will be an important direction for watermark community.
> - **Context-aware semantic-robust random seeds:** Context-aware seeding can alleviate seed dislocation issues. However, as noted in Appendix D, traditional hash-based seeders are still sensitive to paraphrasing. We therefore argue that a semantic-robust random seeder based on the embedding of the previous sentences rather than token hashing should help. Additional experiments in Appendix D.2 confirm the potential of this idea under cross-sentence paraphrasing. We plan to provide more detailed results in the revised version to better illustrate PMark’s adaptability and the challenges of cross-sentence paraphrasing.

---

> ### Comment · Reviewer_zq8u · 2025-11-26
> **Thank you**
>
> Thank you for the response! The additional thoughts are reasonable and interesting to me. I believe this paper deserves my positive assessment.

---

> > ### Author Response · Authors · 2025-11-27
> > **Thank you**
> >
> > Thank you for your constructive feedback and for your recognition of our work!

---

### Author Response · Authors · 2025-12-01
**Global Response and Rebuttal Summary**

Dear Program Chairs, Senior Area Chairs, and Area Chairs,

To summarize, this is the first work to provide a unified theoretical framework for Semantic-level Watermark (SWM) methods, the first distortion-free SWM approach, and the first to highlight and realize dense watermark evidence in SWM.

We sincerely thank all reviewers for their thorough and constructive feedback. We are particularly encouraged by the collective recognition of our work's key strengths, including our innovative and solid theoretical analysis (Reviewers `zq8u`, `SZC6`, `dwYt`), novel and elegant SWM method (Reviewers `zq8u`, `4inb`, `SZC6`, `dwYt`), comprehensive experiments with promising performance (Reviewers `zq8u`, `4inb`, `SZC6`, `dwYt`), and clear presentation (Reviewers `zq8u`, `SZC6`).

In our detailed rebuttals, **we have endeavored to address every concern raised with new theoretical analysis and quantitative results**. This includes additional analysis on offline/multi-channel settings (Reviewers `4inb`, `SZC6`), solutions to cross-sentence attacks (Reviewers `zq8u`, `4inb`, `SZC6`) with additional experimental support, and analysis regarding computational overhead and potential solutions (Reviewers `4inb`, `dwYt`).

We are glad that our response was recognized by Reviewer `zq8u` on Nov 26 17:59 UTC. While we regret that we didn’t have the opportunity to exchange further insights with reviewers, we deeply appreciate their professional and insightful feedback, which has significantly improved our manuscript and deepened our understanding.

We hope our work contributes meaningfully to SWM research, and wish the best with our community.

Best,
Submission 7258 Authors

---

### Meta-Review · Area_Chair_qy3A · 2026-01-06

**Summary:**

This paper introduces PMark, a semantic-level watermarking method for large language models that is both distortion-free and robust against paraphrasing attacks. The authors provide a unified theoretical framework using proxy functions and propose a multi-channel sampling approach to strengthen watermark evidence. Experiments show PMark outperforms existing methods in both text quality and robustness, offering a practical and theoretically sound solution for detecting AI-generated text.

**Reviewer Concerns:**

Most concerns raised by reviewers were addressed in the rebuttal. The authors clarified theoretical guarantees for multi-channel robustness, provided additional experiments for cross-sentence attacks, and discussed computational overhead and offline variants. Minor issues remain, such as deeper theoretical analysis for offline mode and practical guidance for hyperparameters, but these do not significantly affect the overall contribution.

**Reviewer Scores:**

+ Reviewer zq8u: Already positive (rating 8). Likely unchanged after rebuttal.
+ Reviewer SZC6: Positive (rating 8). No major concerns remain; score likely unchanged.
+ Reviewer 4inb: Initially marginally above acceptance (rating 6). Rebuttal addressed key questions; score could increase slightly.
+ Reviewer dwYt: Initially marginally above acceptance (rating 6). Concerns on efficiency were acknowledged; score likely unchanged or slightly higher.

---

### Decision · Program_Chairs · 2026-01-26

Accept (Poster)